# Tests as Instructions: A Test-Driven-Development Benchmark for LLM Code Generation

## Abstract

This paper focuses on test-driven development (TDD) tasks, where test cases act as both instruction and verification for LLM code generation. We build a TDD benchmark to evaluate frontier models, where reasoning models of OpenAI achieve SOTA. We identify instruction following and in-context learning as the critical abilities for all models to succeed at TDD tasks. We further reveal their vulnerabilities to long instructions as an area of improvement.

## 1 Introduction

Code generation/completion is a classical LLM (large language model) task and one of its most important applications. The focus of current research and benchmarks include algorithms(Austin et al., 2021; Chen et al., 2021), tool use(Yan et al., 2024), debugging(Jimenez et al., 2024), etc. Here, a coding task consists of two parts: formulation and verification. The formulation is the task description in natural language, accompanied with code snippets when necessary. The verification is a collection of tests to run against the code generated by LLM. If all tests pass, the evaluated LLM is considered to succeed at the task.

In this paper, we explore a new avenue where tests are *both* task formulation and verification. Specifically, a number of tests in its original code format are copied and pasted into the LLM prompt. A simple prefix is attached instructing the LLM to generate code to pass all given tests. The generated output is evaluated by the same tests. We call this a TDD (test-driven development) task, and for the evaluation purpose, an essemble of TDD tasks a TDD benchmark.

Using the running example in (Beck, 2022), Tab. 1 illustrates how a multi-currency conversion application is gradually built via TDD. Each row represents a feature request codified as a test. Human developers approach TDD *incrementally*, one row per iteration. In each iteration, they write new code to pass the current test, as well as all tests in previous iterations. LLMs approach TDD *at once*. As demonstrated in this paper, they are able to write code to pass multiple tests in one inference, i.e. aggregate multiple iterations into one batch.

Also wordings in the **TDD Task** column are akin to natural language instructions in classical coding benchmarks. For human developers, these are considered best practices to bring clarity to their coding work. On the other hand, LLMs are able to infer them from tests included in the prompt. As such, one can let LLMs skip this step to directly output code, or ask them to explicitly spell out the task via chain of thoughts.

We argue that TDD is the norm of modern application development across many domains, from enterprises to startups, consuming numerous engineering hours. In these production environments, tests are the de facto system specs overriding documentations. The maturity of a system can be measured by the size and age gap of its test suite. As such, LLMs with strong TDD capabilities would generate tremendous value to the software industry.

Since tests play pivotal roles in all coding benchmarks, we use Tab. 2 to compare them in three categories. The key differentiator between algorithmic and TDD benchmarks is the semantic mutability, where tests evaluate solutions to the problem in the former, but define the problem itself in the latter. The key differentiator between problem-solving and TDD benchmarks is the scope, where tests are

Table 1: An example of test-driven development tasks

| Test Name | Verifications | Feature | TDD Task |
|---|---|---|---|
| testMultiplication | Ensures multiplying a dollar amount by an integer gives correct results | Basic dollar multiplication | Defines a Dollar class with amount and times() method |
| testEquality | Confirms that two Money objects are equal if they have the same amount and currency | Equality check for Money objects | Adds an equals() method for comparing amounts |
| testFrancMultiplication | Verifies multiplication functionality for francs | Introduction of Francs | Creates a Franc class similar to Dollar with multiplication |
| testCurrency | Checks that each Money object correctly identifies its currency | Currency attribute | Adds a currency attribute in Money class |
| testSimpleAddition | Tests addition of two Money objects within the same currency | Simple addition of Money objects | Implements plus() for adding same-currency Money objects |
| testIdentityRate | Validates that the Bank provides a 1:1 exchange rate for the same currency | Bank class for currency conversion | Introduces Bank class with exchange rates |
| testReduceMoney | Ensures correct conversion of sums between different currencies | Addition with conversion | Implements conversion of sums across currencies in Bank |
| testMixedAddition | Verifies handling of mixed currency addition and conversion to target currency | Complex expressions with different currencies | Refactors to handle mixed-currency expressions with conversions |

Table 2: Comparison of coding benchmarks

| | Algorithmic Benchmarks | TDD Benchmarks | Problem-Solving Benchmarks |
|---|---|---|---|
| **Example** | Palindrome Check in HumanEval(Chen et al., 2021) and LeetCode | Multi-currency conversion in Tab. 1 | scikit-learn-14520 issue in SWE-bench (examplified in (OpenAI, 2024)) |
| **Semantic Mutability** | Immutable to test changes | Always mutable to test changes as a form of feature request | Sometimes mutable to test changes as a form of verification |
| **Scope** | Function | Module | Application |
| **Context Length Requirement** | Short | Medium | Long |
| **Running Overhead** | Low (programming language dependencies) | Medium (programming language and framework dependencies) | High (containerized dependencies) |
| **Source of Pre-train Knowledge** | Textbooks and online documents | Open source repos | Open source repos |
| **Primary Applications** | Coding interview and brainstorming | Pre-launch feature development | Post-launch patch and bugfix |

only a slice of the context (along with logs, source code, and the issue description) in the former, but solely comprises the entire context in the latter.

Below are original contributions of this paper.

- **TDD Tasks and Benchmarks**: We propose the definition of TDD tasks and built to our knowledge the first TDD benchmark. This effort reveals the following insights.

- **Critical Abilities for TDD Task Success**: Pre-train coding knowledge is necessary but insufficient for the success of TDD tasks. We identify the following critical abilities: instruction following, in-context learning, and reasoning. We expect this list to grow as the investigation deepens.

- **More tests cause worse performance**: The SOTA is significantly lowered after more tests are added, which limits the application scope of LLM-driven TDD. The suspect root cause for the performance bottleneck, e.g., attention decay(Liu et al., 2024), is unconfirmed.

The rest of this paper is organized as follows. In Sec. 2, we introduce how the benchmark is built and run. In Sec. 3, we show LLM performance and analyze their errors. In Sec. 4, we discuss the degraded LLM performance under more test cases, and potential root causes. Finally, we discuss related works in Sec. 5 and conclude the paper in Sec. 6.

## 2 BENCHMARK

### 2.1 SCOPE

We envision the flourishing of many affordably-trained coding LLMs or SLMs (small language models)(Li et al., 2023; Lozhkov et al., 2024; Hui et al., 2024; Huang et al., 2024) with specialties scoped by the template below.

$$\{\text{Programming Language}, \text{Framework}, \text{Domain}, \text{Task}\}$$

Given a software engineering project, the first three elements are decided before the project begins, and do not change during the project. For the last element (task), the engineer regularly shuffles among several tasks, such as comment generation, test generation, code generation, code interpretation. As the above template has many instances, we argue for the need of many benchmarks, one for each instance. As a start, this paper presents a benchmark for the following instance.

$$\{\text{JavaScript}, \text{React}, \text{Web App}, \text{TDD Code Generation}\}$$

In addition, our choice is backed by following considerations.

- **Representation** Founded in 2013, React(Meta, 2013) is one of the most popular open source projects and a top choice for web app developers. High-quality React code is abundant in any LLM pretraining corpuses. Hence, one can safely assume a general or coder LLM to have sufficient React knowledge to complete the task without seeking exernal knowledge.

- **Scalability** As demonstrated in Tab. 3, React code is succinct and compositional, able to implement diverse functionalities in one file. This makes it very convenient for us to innovate evaluations. First, we can define comprehensive tasks even if the LLM has limited context window. Second, by varying test suite size, we can scale the instruction scope to search for model limits and hill-climbing directions.

- **Ecosystem** The rich and stable ecosystem of React blesses us with many excellent tools. They greatly facilitates the evaluation work especially the test verification.

### 2.2 TASK FORMULATION

Since the formulation of a TDD task primarily consists of verbatim test code, we use a sample web app scenario to explain.

Consider a blogging website, in which a user adds comment to an existing blog post. This user journey is simulated by the unit test in Table 4. Here, `fetchMock.post` is a lightweight setup to

Table 3: Template of React-based solution

```
// Import Statements
...
import React from 'react';

// Main component of the application
function App() {
 ...
 // Business logics to handle user actions
 const functionA = (...) -> {
  ...
 };
 const functionB = (...) -> {
  ...
 };

 // JSX-based UI layout
 return (
  <div>
   // UI events are wired to the calling of functionA and functionB
  </div>
 );
};

// Export Statement
export default App;
```

mock a successful API response without running any additional software components. The following `await` lines simulate user actions (text input, mouse click etc.). Finally, `expect` lines examine the expected outcome, i.e. the mocked API should be invoked exactly once and the system response of success should appear on the updated webpage. Similarly, the pairing failure case is shown in Table 5, where a mocked API failure is expected to lead to error message on the updated webpage.

Table 4: Success case for adding a comment to a blog post

```
test("successfully adds a comment to a post", async () => {
  fetchMock.post("/api/comments", 200);

  await act(async () => {
    render(<MemoryRouter><App /></MemoryRouter>);
  });
  await act(async () => {
    fireEvent.change(screen.getByPlaceholderText(/Add a comment/i),
    { target: { value: "Great post!" } });
  });
  await act(async () => {
    fireEvent.click(screen.getByText(/Submit/i));
  });

  expect(fetchMock.calls("/api/comments").length).toBe(1);
  expect(screen.getByText(/Comment added successfully/i)).toBeInTheDocument();
}, 10000);
```

Table 5: Failure case for adding a comment to a blog post

```
test("fails to add a comment to a post", async () => {
  fetchMock.post("/api/comments", 500);

  // Lines identical to the success case are ignored.

  expect(screen.getByText(/Failed to add comment/i)).toBeInTheDocument();
}, 10000);
```

The prompt is straightforward: we feed test files to the LLM, expecting it to generate code passing these tests. The token length of the prompt is around 0.5K.

$$\text{Generate App.js to pass the tests below:} \quad (1)$$
$$\{Tab.\ 4\}\{Tab.\ 5\}.\ \text{RETURN CODE ONLY.}$$

The benchmark consists of 1000 such tasks. Each task uses a success case and failure case to describe the scenario. These 1000 tasks are aggregated under 20 applications, e.g. blogging, e-commerce, traveling. More details can be found Appendix A.

## 2.3 TASK VERIFICATION

To succeed at the task defined in Tab. 4 and 5, an LLM is expected to output code following the template in Tab. 3. The code generates a single webpage decorated with a form-like UI element allowing the test-simulated user to add comment. If all expectations in Tab. 4 and 5 are met, the tests pass.

We use $pass@k$, a metric defined in (Chen et al., 2021) and commonly accepted by subsequent works. Due to budget and rate limit constraints, each task is evaluated at most 10 times, i.e. $n = 10$. Since $k$ must be no larger than $n$, we measure $pass@1$, $pass@5$, and $pass@10$. More details on the experiment setup can be found in Appendix B.

## 3 EVALUATION RESULTS

### 3.1 LLM PERFORMANCES

Tab. 6 summarizes the $pass@k$ results of 18 frontier LLMs. We only measure $pass@1$ for o1 models primarily due to their inference cost. But since the value of $pass@k$ asymptotically increases with $k$, there is no doubt that the o1 models lead non-reasoning LLMs by an obvious gap. Also worth noting is the impressive performance of open-source LLMs, with deepseek-v2.5 as the top contender.

Table 6: pass@k results for frontier LLMs

| Model | pass@1 | pass@5 | pass@10 |
|---|---|---|---|
| o1-preview | 0.952 | N/A | N/A |
| o1-mini | 0.939 | N/A | N/A |
| gpt-4o-2024-08-06 | 0.885 | 0.9047 | 0.909 |
| claude-3.5-sonnet | 0.8808 | 0.8845 | 0.886 |
| deepseek-v2.5 | 0.834 | 0.8595 | 0.869 |
| gpt-4o-mini | 0.8271 | 0.8534 | 0.858 |
| mistral-large-2 | 0.7804 | 0.8191 | 0.831 |
| deepseek-coder-v2-instruct | 0.7002 | 0.8009 | 0.827 |
| gemini-1.5-pro | 0.6813 | 0.7678 | 0.795 |
| gemini-1.5-flash | 0.57 | 0.6427 | 0.663 |
| deepseek-coder-v2-lite-instruct | 0.4606 | 0.6144 | 0.653 |
| mixtral-8x22b-instruct | 0.3074 | 0.4821 | 0.533 |
| llama-v3-70b-instruct | 0.3323 | 0.4462 | 0.489 |
| llama-v3p1-405b-instruct | 0.302 | 0.4053 | 0.437 |
| llama-v3p1-8b-instruct | 0.2512 | 0.3941 | 0.432 |
| llama-v3p1-70b-instruct | 0.1027 | 0.1848 | 0.246 |
| mixtral-8x7b-instruct | 0.1269 | 0.196 | 0.218 |
| llama-v3-8b-instruct | 0.0679 | 0.1183 | 0.139 |

## 3.2 BENCHMARK DIFFICULTY

We made each model solve each task for 10 times, which gives us 160 solutions per task[1]. Fig. 1 shows number of failures per task. The more failures a task collects, the more difficult it is.

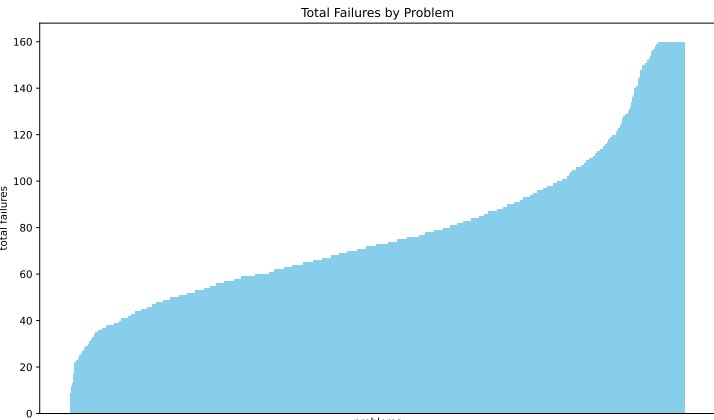

Figure 1: Failures per problem

As indicated by the figure, the majority of the problems have low failure rates, i.e. they are relatively easy for LLMs to solve. Conversely, a small cluster of problems on the far right exhibit extremely high failure rates, some remain unsolved by any model. Appendix E will reveal more insights on why they are difficult.

## 3.3 DOES THE CODE BUILD?

Of the total 160,000 solutions included in Fig. 1, only 172 have syntax errors, i.e. the build failure rate is 0.1%. In particular, the solutions by o1 models, Claude 3.5, and Mistral Large 2 have no syntax errors. We manually examined a subset of built solutions, and found that they support the feature the test cases intend to evaluate.

This means all LLMs are able to follow high-level instructions and write quality code. The real challenge for them is to meet all test expectations, some explicit and others implicit, therefore failing the task. To verify this hypothesis, we ran an alternative experiment following the TLD (test-last development) approach which significantly boosted $pass@1$ of all tested models. Details are in Appendix C.

## 3.4 ERROR TYPES

We study error logs and find LLMs make seven types of errors, coded to A through G. They are summarized in Tab. 7.

The verbatim errors are the original error messages or codes captured by the log. Each of them is broadly scoped to contain a wide array of behaviors. However, in the context of our benchmark, we find all verbatim errors are projected to a narrowband of behaviors attributed to the same root causes.

Based on the root causes, we further conjecture their connections to model abilities.

- **Preference Alignment**: violating unspecified user preference, i.e. the latest stable version
- **In-context Learning**: mistmatching string or integer values specified in the model input
- **Instruction Following**: misunderstanding or missing the feature requested in test cases
- **Pretraining Knowledge**: violating scoping rule of the programming language

---

[1]We exclude o1 models because they are only evaluated once per task. Also given their high success rates, they leave very small impact to the distribution.

Table 7: Error table

| Error Code | Name | Verbatim Error | Root Cause | Model Ability |
|---|---|---|---|---|
| A | Version Mismatch | TypeError | Deprecated framework functions are used | Preference Alignment |
| B | Text Mismatch | TestingLibrary ElementError | Attributes or texts of HTML tags do not match test expectations | In-context Learning |
| C | API Call Mismatch | expect(received) | Mock APIs are called less or more than expected | In-context Learning |
| D | Uninstalled Module | Cannot find module | Imported module is not installed | Instruction Following |
| E | Invalid API Call | fetch-mock | The call signature does not match the test expectation | In-context Learning |
| F | Scope Violation | ReferenceError | An out-of-scope call is made to a locally-defined function | Pretraining knowledge |
| G | Missing UI Element | Element type is invalid | No UI element is defined in the code | Instruction Following |

## 3.5 SINGULAR AND TWIN ERRORS

An error log can contain a combination of many error types, indicating the code is poorly implemented. But this is not the dominant pattern. 93% of error logs contain either a singular error or twin errors. Fig. 2 shows the distribution of singular and twin errors.

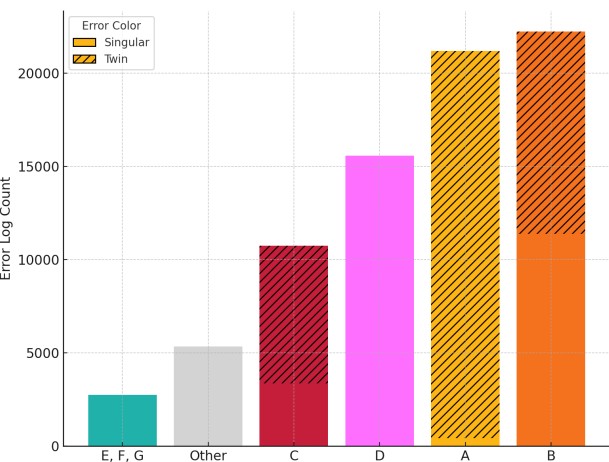

Figure 2: Distribution of singular and twin errors

Singular error means the log contains only one error pointing to a single line. Twin errors are two errors of the same type, preeminently pointing to the same error line. Since the code needs to pass two unit tests, often times the same bug offends both tests. This means that even upon failures, all LLMs produce quality code, but with only one error.

## 3.6 ERROR DISTRIBUTION BY MODELS

In Fig. 3, we show the error distribution separately for each LLM[2]. The most important finding here is that no model is immune to any of the seven error types, even when the raw error counts differ by one order of magnitude bewteen two extremes.

---

[2]o1 models are excluded because their sample sizes are too small (1 run per task instead of 10). They still make the same types of errors as other LLMs.

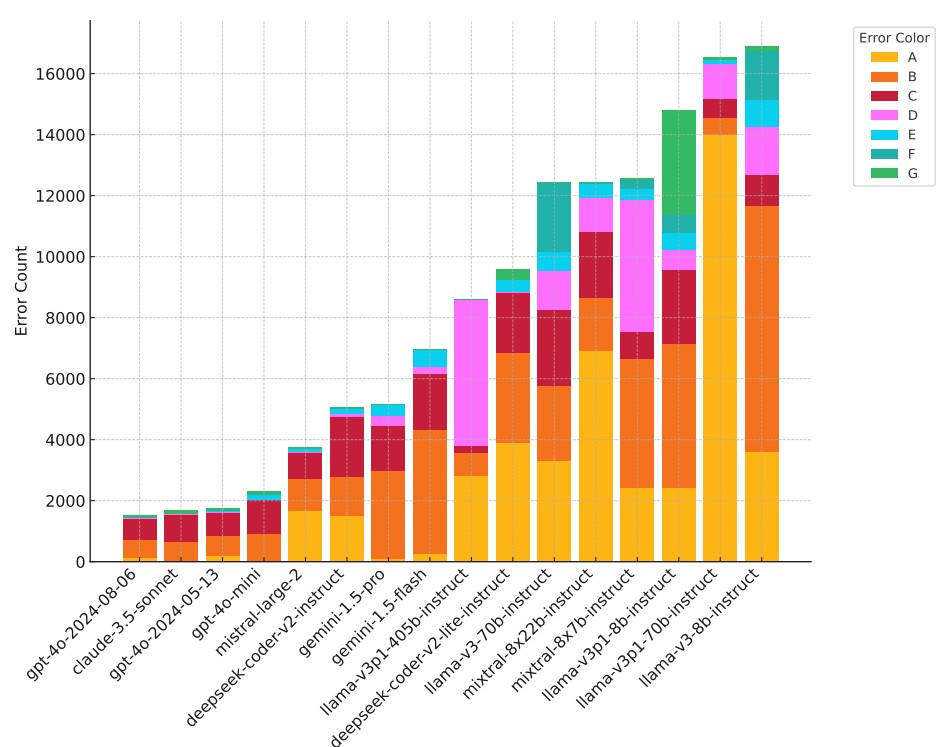

Figure 3: Error distribution by models

This means that all models possess the same knowledge and capabilities to write high-quality code which meets test expectations, and same inherent vulnerabilities resulting in the same types of errors. But top models distinguish themselves at lower error rates, i.e. ability to make fewer errors .

## 4 DUO-FEATURE BENCHMARK

In light of o1 models' superb performance to saturate the benchmark, we propose a more challenging benchmark by merging two TDD tasks into a duo-feature task. Under this new benchmark, each task consists of four test cases: two successes and two failures. Accordingly, the prompt length is doubled to around 1K tokens. Also the output code follows the same template (Tab. 3), and generates a single webpage decorated with multiple UI elements to support two features.

### 4.1 LLM PERFORMNACES

As shown in Tab. 8, more test cases cause $pass@1$ of all LLMs to decrease significantly. Also the SOTA is owned by Claude 3.5.

Table 8: Duo-feature benchmark: pass@1 for selected LLMs

| Model | pass@1 |
|---|---|
| claude-3-5-sonnet | 0.679 |
| o1-mini | 0.667 |
| o1-preview | 0.652 |
| gpt-4o-2024-08-06 | 0.531 |
| deepseek-v2.5 | 0.49 |
| mistral-large-2 | 0.449 |

However, the model behaviors remain largely the same on other aspects described in Sec. 3. The output code is functional with occasional build failures, and make the same errors more frequently.

## 4.2 INSTRUCTION LOSS

To study why o1 models perform worse than Claude 3.5, we find a task solved by Claude 3.5, but failed by o1-preview. As shown in Tab. 9, this task requires the duo feature of adding comment and retrieving blog posts in a single webpage.

Table 9: A duo-feature TDD task: add comment and retrieve all blog posts

```
import App from './addComment_retrieveAllBlogPosts';
...
test('successfully adds a comment to a post', async () => {
... }

test('fails to add a comment to a post', async () => {
... }

test('Success: retrieve a list of all blog posts', async () => {
... }

test('Failure: retrieve a list of blog posts with server error', async () => {
  fetchMock.get('/api/posts', { status: 500, body: { error: 'Internal Server Error' } });
  ...
  expect(fetchMock.calls()).toHaveLength(1);
  expect(screen.getByText('Internal Server Error')).toBeInTheDocument();
}, 10000);
```

Here, o1-preview passes all tests but the last one. The output code neither attempts to catch the 500 error nor prints out the **Internal Server Error** string. The reasoning chain is normal, and no step specifically mentions the need to catch internal server errors.

Crafting the component $\longrightarrow$ Laying out the requirements $\longrightarrow$
Importing dependencies $\longrightarrow$ Breaking down the code $\longrightarrow$
Setting up the app $\longrightarrow$ Testing a post functionality $\longrightarrow$
Testing API integration

The o1-preview's inherent coding ability is solid, because it solves both tasks separately under the single-feature benchmark. To this end, we suspect the root cause to be instruction loss. It remains unknown whether the instruction is never picked up from the model input, or lost during an early reasoning stage. What we are sure of is the necessiry of full instruction set as the foundation for reasoning, without which the reasoning model will simply fail the task.

## 5 RELATED WORKS

### 5.1 CODING-RELATED TASKS AND BENCHMARKS

Prompt-driven coding has become mainstream since the introduction of Codex(Chen et al., 2021). The evolution of benchmarks reflect the scaled-up challenges posed to LLMs, from algorithms(Austin et al., 2021), to data science problems(Lai et al., 2022), object-oriented coding(Du et al., 2023), code execution(Yu et al., 2023), function calling(Yan et al., 2024), SQL queries(Gao et al., 2023), project-level resolution(Jimenez et al., 2024), etc. These benchmarks all rely on test suite of different sizes to verify task success. On the other hand, the task formulation, i.e. the prompt, is becoming longer and harder to specify, resulting in misalignment with its verification counterpart, which can be only addressed by human calibration(OpenAI, 2024).

TDD benchmarks avoid such misalignment by unifying task formulation and verification, meanwhile introducing other challenges to LLMs.

## 5.2 INSTRUCTION FOLLOWING AND IN-CONTEXT LEARNING

Instruction following and in-context learning are two of the most desired LLM abilities to ace TDD tasks. Both topics have been extensively researched(Dong et al., 2024; Lou et al., 2024), and their close relations revealed by several empirical or mechanistic studies(Wei et al., 2022; Li et al., 2024; Xie et al., 2022; Hewitt et al., 2024; Singh et al., 2024). Several well-known benchmarks(Chia et al., 2023; Jiang et al., 2024; Qin et al., 2024) were also introduced to measure LLM progress on these abilities.

However, majority of the existing works focus on natural language instructions. Given the practical values of TDD tasks, we would like to see more interests developed over code-based instructions. Our evaluation demonstrates LLMs' remarkable ability to follow coded instructions. But it also revealed their vulnerabilities when coded instructions grow longer. This is related to another stream of works which try to scale natural language instructions(Son et al., 2024; Cheng et al., 2023). We will track closely the development of these two work streams.

## 5.3 REINFORCEMENT LEARNING AND REASONING

The o1 models have been speculated to leverage many seminal works on reinforcement learning and reasoning. Works on the learning side include self-play(Zhang et al., 2024), self-taught(Zelikman et al., 2022; 2024), learning from running environment(Silver et al., 2017), etc. Works on the inference side include process modeling(Lightman et al., 2023), inductive reasoning(Wang et al., 2024), tree search(Anthony et al., 2017), etc.

Aside from general reasoning models like o1, many works have applied reinforcement learning to coding-specific problems, including code generation(Jain et al., 2023), test generation(Steenhoek et al., 2023), error repair(Islam et al., 2024b;a), etc.

The values of reasoning and self-improvement techniques to TDD tasks are best showcased by the exciting SOTA lift to our benchmark. Unfortunately, we also observe the negative impact of instruction loss to reasoning model performances. We think it is worthwhile to incorporate nuanced and complex model input into future reasoning model development.

## 5.4 TDD IN LLM CODING

Much similar to this paper, some recent works introduced TDD to coding task formulation, and studied best practice and performance impact(Mathews & Nagappan, 2024; Murr et al., 2023; Piya & Sullivan, 2023). But to our knowledge, this is the first paper focusing on TDD benchmarking.

Finally, one may argue that it is easy to repurpose classical coding benchmarks to evaluate TDD tasks by simply appending their test cases to the prompt. But we argue the benefits and necessity to have dedicated benchmarks to this cause. Just as TDD is the norm in application development emphasizing on business logic, knowledge on input instructions is the most critical factor to task success, overshadowing pretraining knowledge[3]. We think benchmarks crafted along this line of thinking can appropriately evaluate and challenge LLMs to keep improving on TDD tasks.

## 6 CONCLUSIONS

This paper focuses on the TDD aspect of LLM code generation, and claims two contributions. The first is a dedicated TDD benchmark which we use to evaluate 18 frontier LLMs. The second is the insights obtained via the evaluation. Specifically, instruction following and in-context learning are the key areas of improvement for LLMs and reasoning models to excel on more challenging TDD tasks.

There are two future directions. The first is to grow our benchmark to cover more application scenarios, meanwhile cross-examining learnings from this paper. The second is to explore practical hill-climbing ideas to address the vulnerability to long coded instructions.

---

[3]This is a comparative argument relevant to other tasks akin to algorithms and data structures. A TDD task cannot succeed without a strong coding LLM.

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

## A    BENCHMARK CONSTRUCTION

The construction of the benchmark follows the methodology of Self-Instruct(Wang et al., 2023). As the initial step, humans proposed 20 web applications listed in Tab. 10, referencing main applications of JavaScript and React(Accomazzo et al., 2017; Mozilla, 2005; fir, 2017).

Subsequently, five categories are proposed for each application, shown in Tab. 11. Using these human-generated seeds, we prompt GPT-4o to propose, for each category, 10 scenarios, each described by a sentence. This results in a total of 1000 scenarios for the benchmark. As the final step, after reviewing these scenarios by humans, we further GPT-4o to generate a success test and failure test for each scenario, exemplified in Sec. 2.2.

## B    EXPERIMENT SETUP

The most straightforward way for us to access LLMs are public token-based APIs. For top close-sourced models, our only option is via the owners' APIs. The top open-sourced models are hosted by a few platforms, among which we choose Fireworks.

Although each API bears its minor difference, all APIs are heavily influenced by the design of OpenAI API. Tab. 12 lists the tunable parameters exposed by each API. Since we do not know the default parameter value set by each API provider, we explicitly set the same parameter values to all LLMs under evaluation, whenever applicable. To limit the search space, we only tune $temperature$ and $top\_p$, the two most popular parameters available on all platforms. For other parameters, we assign fixed value to them across all LLMs.

We conducted a grid search to locate a sweet spot at which all LLMs deliver near-best results. We chose 100 random problems from the benchmark, 5 out of each application. We then choose the large model out of the five leading model families, and measured their $pass@1$ ($n = 1$) on the discrete 2D space of $temperature$ and $top\_n$, where $temperature = 0, 0.1, 0.2, ..., 1$, and $top\_p = 0, 0.1, 0.2, ..., 1$.

Tab. 13 presents the lowest and highest $pass@1$ value by each LLM in this grid search. Based on the results, we finalize our parameters as follows.

$$temperature = 0.2$$
$$top\_p = 0.8$$
$$top\_k = 40$$
$$presence\_penalty = 0$$
$$frequency\_penalty = 0$$

Results of our full-scale evaluations also align with this small-scale experiment, except for the Deepseek coder model whose performance exceeds expectation. Also worth noting is that open-source models exhibit larger performance variation than closed-source models.

Table 10: Applications of the benchmark

| Name | Overview |
|---|---|
| blogging | A content management system for creating and managing blogs, with features like user registration, post creation, categorization, commenting, and SEO optimization. |
| customer support | A help desk application where users can submit support tickets, track their status, access a knowledge base, and chat with support agents. |
| e-commerce | A fully functional e-commerce site with features like product listings, shopping cart, user authentication, order processing, and payment integration. |
| event management | An app for organizing events, including event creation, ticket sales, attendee registration, and scheduling |
| fitness tracking | An application for tracking fitness activities, setting goals, monitoring progress, and integrating with wearable devices. |
| inventory management | A web application designed to help businesses track and manage their inventory. Features include product cataloging, stock level monitoring, automated reorder alerts, supplier management, sales and purchase order processing, and detailed reporting on inventory performance. |
| job board | A job listing site where employers can post job openings and job seekers can search and apply for jobs. |
| music streaming | A platform for streaming music, creating playlists, and discovering new artists. |
| news aggregator | A news platform that aggregates articles from various sources, categorizes them, and allows users to customize their news feed. |
| online learning | An LMS where users can enroll in courses, watch videos, complete quizzes, track progress, and receive certificates. |
| online marketplace | A platform for buying and selling goods, similar to eBay, with features like user ratings, bidding, and secure transactions. |
| personal finance | A tool for managing personal finances, including expense tracking, budget planning, report generation, and financial goal setting. |
| pet care | a web application designed to help pet owners maintain a detailed record of their pet's health, activities, and milestones. |
| photo gallery | An application for uploading, organizing, and sharing photos, with features like tagging, album creation, and social sharing. |
| real estate | A platform for listing and searching real estate properties, with features like property details, image galleries, map integration, and contact forms. |
| recipe sharing | A platform where users can share, search, and save recipes, with features like ingredient lists, cooking instructions, and user ratings. |
| social media | A social media platform where users can create profiles, post updates, follow others, like and comment on posts, and manage a feed of updates. |
| task management | An application for managing tasks and projects, with features like task creation, assignment, progress tracking, and notifications. |
| travel planning | An app for planning and booking travel, including flight and hotel searches, itinerary creation, and travel recommendations |
| weather | An app that provides real-time weather updates, forecasts, and severe weather alerts. |

## C  A TEST-LAST DEVELOPMENT (TLD) EXPERIMENT

Of all error types in Tab. 7, type A, B, and D account for overwhelming share among LLMs with weaker performances (Fig. 3). On the other hand, these errors do not indicate the code is dysfunctional, only violating the tests. In light of this counter argument, we conducte a TLD (test-last development) experiment, where we rewrite the violated tests to accommodate the verbtaim code output of these models.

- **Type A Error** Rollback to an older version of React if the code uses functions therein

- **Type B Error** Retrofit attribute or text property expectations to match the code

- **Type D Error** Refactor mock statements to accommodate the module referenced in the code

Table 11: Categories for each application of the benchmark

| Name | Categories |
|---|---|
| blogging | Post Management, Categorization and Tag Management, Commenting System, SEO Optimization, Post Analytics |
| customersupport | Ticket Management, Agent and Collaboration, Knowledge Base, Notifications and Automation, Reporting and Analytics |
| ecommerce | Product Listings, Shopping Cart, Order Processing, Payment Integration, Product Reviews |
| eventmanagement | Event Creation, Ticket Sales, Attendee Registration, Scheduling, General Event Management |
| fitnesstracking | Activity Management, Goal Setting and Tracking, Progress Monitoring, Health and Nutrition, Device Integration and Data Management |
| inventorymanagement | Product Cataloging, Stock Level Monitoring, Supplier Management, Order Processing, Reporting |
| jobboard | Job Posting Management, Job Search and Viewing, Job Application Process, Employer Application Management, User and Profile Management |
| musicstreaming | Search and Discovery, Playback Control, Playlist Management, User Interaction, Advanced Features |
| newsaggregator | Article Management, User Preferences, Article Interactions, Content Customization, User Engagement |
| onlinelearning | Enrollment and Progress Tracking, Course Content and Interaction, Assessment and Certification, User Interaction and Communication, Course and Content Management |
| onlinemarketplace | Product Management, Checkout and Payment, Order Management, Search and Navigation, Bidding and Auctions |
| personalfinance | Expense Management, Income Management, Budget Planning, Report Generation, Financial Goal Setting |
| petcare | Pet Profiles, Daily Activities, Health Tracking, Reminders, Community |
| photogallery | Photo Upload and Management, Photo Tagging and Organization, Photo and Album Sharing, Photo Interaction and Social Features, Advanced Photo Features |
| realestate | Search and Filters, Sorting and Viewing, User Interaction, Property Management, Additional Features |
| recipesharing | Recipe Management, Search and Filtering, User Interactions, Recipe Viewing, User Profiles and Preferences |
| socialmedia | Profile Management, Post Management, User Interactions, Notifications, Feed Management |
| taskmanagement | Task Management, Project Management, User Management, Task Tracking, Advanced Features |
| travelplanning | Flight Search and Booking, Hotel Search and Booking, Itinerary Creation, Travel Recommendations, General Booking Logic |
| weather | Current Weather Data Retrieval, Weather Forecast Retrieval, Severe Weather Alerts, Location-based Services, User Preferences and Settings |

Table 12: Tunable parameters on different APIs

| | temperature | top_p | top_k | presence_penalty | frequency_penalty |
|---|---|---|---|---|---|
| GPT4o | Y | Y | N | Y | Y |
| Claude | Y | Y | Y | N | N |
| Gemini | Y | Y | Y | N | N |
| Fireworks | Y | Y | Y | Y | Y |

To prevent test semantic drifts, we ensure that the test code structure is unmodified, and restrict each of the above actions to the scope of single statement. As shown in Tab. 14, all LLMs demonstrate significant $pass@1$ lift after test modification.

Table 13: Parameter tuning results on pass@1

| Model | Lowest | Chosen (temperature = 0.2, top_p = 0.8) | Highest |
|---|---|---|---|
| gpt-4o | 0.81 | **0.88** | 0.9 |
| claude-3.5-sonnet | 0.82 | **0.85** | 0.86 |
| deepsseek-coder-v2-instruct | 0.42 | **0.59** | 0.59 |
| gemini-1.5-pro | 0.59 | **0.65** | 0.69 |
| llama-v3-70b-instruct | 0.19 | **0.31** | 0.34 |

Table 14: TLD experiment: pass@1 results

| Model | TDD pass@1 | TLD pass@1 |
|---|---|---|
| llama-v3-70b-instruct | 0.3323 | 0.6400 |
| mixtral-8x22b-instruct | 0.3074 | 0.8000 |
| llama-v3p1-405b-instruct | 0.3020 | 0.8850 |
| llama-v3p1-8b-instruct | 0.2512 | 0.7550 |
| mixtral-8x7b-instruct | 0.1269 | 0.7300 |
| llama-v3p1-70b-instruct | 0.1027 | 0.7900 |
| llama-v3-8b-instruct | 0.0679 | 0.6500 |

Note that TLD is a popular approach for experimental and prototyping projects, but is widely considered a malpractice for high-stake projects. Also TLD bears an implicit cost, since the work test modification itself is also time-consuming.

## D  PROMPT EXPERIMENTS

We also study whether more sophisticated prompts can lift the model performance.

The first experiment is ***system prompt***, which assigns an explicit role to the LLM and raises its awareness. Available in all APIs we run, it complements the user prompt (Equation (1)) which gives detailed instructions to LLM. Equation (2) shows our system prompt.

$$\text{You are a code generator.} \tag{2}$$

The second experiment is ***verbose comment***, which aims to help LLMs better understand the semantics of tests it tries to pass. For each of the 1000 problems, we feed its test code to GPT-4o and ask for English summary of the expectation in multiple sentences. The summary is then inserted into the test code. Tab. 15 shows the verbose comment variant of the test code in Tab. 4.

Table 15: Verbose cmment variant of the test case in Tab. 4

```
test(
"This test case verifies that a comment can be successfully added to a post by simulating
a successful POST request to the '/api/comments' endpoint. The test ensures that the
API call occurs exactly once and that a success message ('Comment added successfully')
is displayed upon successful submission. This helps confirm the correct interaction
between the frontend and backend components when adding comments.",
  async () => {

  // Lines identical to the original test case are ignored.

}, 10000);
```

The third experiment is ***error debugging***. If the generated code fails the test, we add the failed code and the error log to the prompt, hoping the LLM will generate the correct code by learning from its

own mistakes. Below is the prompt.

> $\{failed\_implementation\}$
>
> The above code is the implementation of $\{file\_name\}$. It failed the tests below
>
> $\{success\_test\_code\}\{failure\_test\_code\}$
>
> Below is the test log
>
> $\{error\_log\}$
>
> Try to generate $\{file\_name\}$ again to pass the tests. RETURN CODE ONLY.

For all three prompt variants, we measure $pass@1$ ($n = 1$) against all 1000 problems of the benchmark. Also in each experiment, we apply one prompt variant only, and compare it against the control test using the original prompt (Equation (1)). Tab. 16 summarizes the relative performance gains/loss of each variant.

Table 16: Prompt experiments: pass@1 gain/loss

|  | System Prompt | Verbose Comment | Error Debugging |
|---|---|---|---|
| gpt-4o | -1.3% | -4% | -56% |
| claude-3.5-sonnet | 6.3% | -1% | 38% |
| deepsseek-coder-v2-instruct | -18.2% | 7.5% | -79% |
| gemini-1.5-pro | 6.3% | 2% | 22% |
| llama-v3-70b-instruct | 8.5% | -7.7% | 111% |

To our surprise, we are unable to find a prompt variant delivering universally positive (or negative) impacts to all LLMs. Also we observe the huge swing in the error debugging column. The situation is unique here because this technique is not needed if the model output is correct on the first try. Strong LLMs like GPT-4o can produce high $pass@1$ ($n = 1$) closed to 0.9, which significantly shrinks the sample size.

As such, we can not recommend LLM users to adopt or avoid any prompting technique we have experimented.

## E DEEP DIVES TO O1 MODELS

### E.1 SINGLE-FEATURE BENCHMARK

We deep dive into *ticketSubmission* problem under the *Customer Support* category. The o1 models solved this challenge, which all other LLMs failed. is the. Tab. 17, lists the key steps of the test setup and expectations. We blacken the step which trapped non-reasoning models.

Table 17: ticketSubmission problem

```
test('shows error when submitting a ticket with missing fields', async () =>
  fetchMock.post('/api/tickets',  status: 400 );
  ...
  fireEvent.click(screen.getByText('Submit'));
  ...
  expect(fetchMock.calls('/api/tickets').length).toBe(1);
  expect(screen.getByText('Title is required')).toBeInTheDocument();
, 10000);
```

Similar to all test cases, the mocked API is first setup, followed by simulated user action, then expectations on API access and error message. Non-reasoning models understand the semantics, write functioning code, but fail expectations. The root cause here is the string *Title is required*, which is akin to a technique not requiring API access, aka frontend validation. As a best practice (hence prevelance in pretraining dataset), frontend valiation is lightweight and fast, therefore preferred over backend validation, as shown in Fig. 4. As such, all non-reasoning models are misled to implement frontend validation instead of expected behaviors which is backend validation.

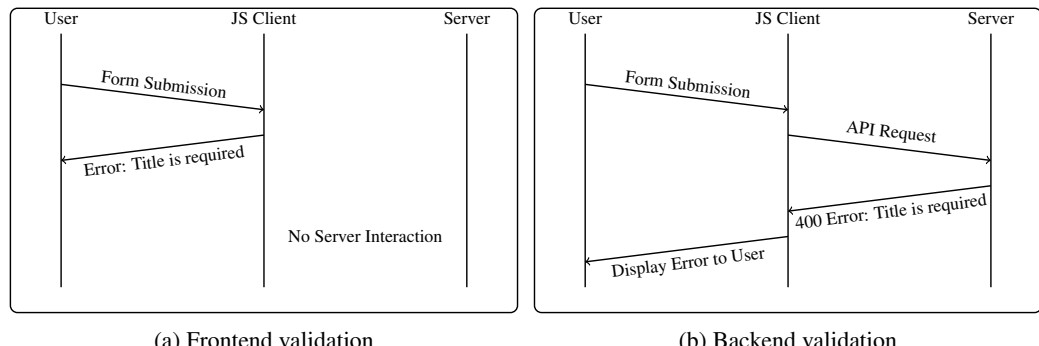

(a) Frontend validation           (b) Backend validation

Figure 4: Comparison of frontend and backend validation

On the other hand, o1 models discover the unpopular yet correct implementation: unconditionally visit the API, and output the *Title is required* error message upon a 400 response. Below is the ChatGPT reasoning chain, in which steps reasoning the 400 response is blackened.

> Mapping out the component $\longrightarrow$ Setting up event handlers $\longrightarrow$
> Setting up the form $\longrightarrow$ Writing test cases $\longrightarrow$
> **Refining the approach** $\longrightarrow$ **Refining error handling** $\longrightarrow$
> **Adjusting error handling** $\longrightarrow$ Adjusting code logic $\longrightarrow$
> Updating JavaScript code

The most crucial step here is **Refining the approach**. Below is its detailed wording.

> I'm updating the code to ensure a fetch request is **always** sent, even without a title.
> The server will respond with a 400 status if the title is absent.

Evidently, the step before it (Writing test cases) conducted certain verification, which leads the model to pivot to the right path.

Unfortunately the reasoning models can also fall for the same trap. Below is a ChatGPT reasoning chain leading o1-preview to the faulty implementation like previous models.

> Mapping out test strategy $\longrightarrow$ Setting up the test $\longrightarrow$
> **Customer service improvement** $\longrightarrow$ Setting up for data $\longrightarrow$
> Setting up the form $\longrightarrow$ **Verifying form submission** $\longrightarrow$
> SHOWING ERRORS $\longrightarrow$ Refining the form handling

On a closer look, step **Customer service improvement** derails the model from backend validation to frontend validation.

> I'm thinking about creating a TicketSubmission component with
> a 'Title' input and 'Submit' button. Submitting the form will trigger
> a POST request to '/api/tickets', validating the 'Title' field **before** submission.

More interestingly, the step **Verifying form submission** does not correct the wrong direction, but solidify it.

> I'm thinking about how the form ensures 'Title' must be filled.
> It sends a POST request **if** 'Title' is entered, showing success
> or 'Title is required' based on the response status.

With these superficial clues, we speculate that the derailing is due to preemption of original expectations by model's inherent knowledge. The subsequent verification step is derived from neighboring steps already derailed, instead of orginal expectations only accessible from the input tokens.

### E.2 DUO-FEATURE BENCHMARK

The duo-feature benchmark was composed in two ways. The first way is shown in Tab. 18 (a), in which the original export name of the single-feature benchmark is preserved as is. The second way is shown in Tab. 18 (b), where the export names are normalized to a unified name **App**.

Table 18: Two formats of the duo-feature benchmark

```
...
import TaskA from './TaskA_B';
import TaskB from './TaskA_B';

test("Success at task A", async () =>
  ...
  render(
    <MemoryRouter><TaskA /></MemoryRouter>
  );
  ...
, 10000);

test("Failure at task A", async () =>
  ...
  render(
    <MemoryRouter><TaskA /></MemoryRouter>
  );
  ...
, 10000);

test("Success at task B", async () =>
  ...
  render(
    <MemoryRouter><TaskB /></MemoryRouter>
  );
  ...
, 10000);

test("Failure at task B", async () =>
  ...
  render(
    <MemoryRouter><TaskB /></MemoryRouter>
  );
  ...
, 10000);
```

(a) Raw format

```
...
...
import App from './TaskA_B';

test("Success at task A", async () =>
  ...
  render(
    <MemoryRouter><App /></MemoryRouter>
  );
  ...
, 10000);

test("Failure at task A", async () =>
  ...
  render(
    <MemoryRouter><App /></MemoryRouter>
  );
  ...
, 10000);

test("Success at task B", async () =>
  ...
  render(
    <MemoryRouter><App /></MemoryRouter>
  );
  ...
, 10000);

test("Failure at task B", async () =>
  ...
  render(
    <MemoryRouter><App /></MemoryRouter>
  );
  ...
, 10000);
```

(b) Normalized format

Tab. 8 shows results from the normalized format. Under the raw format, all models struggle. Most strikingly, o1 models fail all problems (Tab. 19).

Table 19: Duo-feature benchmark raw format: pass@1 results for selected models

| Model | pass@1 |
|---|---|
| claude-3-5-sonnet | 0.32 |
| gpt-4o-2024-08-06 | 0.026 |
| deepseek-v2.5 | 0.02 |
| mistral-large-2 | 0.02 |
| o1-mini | 0 |
| o1-preview | 0 |

To find the root cause, we find the raw format (Tab. 18 (a)) has two imports of different names, i.e. **TaskA** and **TaskB**. But they are actually default imports (without curly braces) which are name-agnostic. Also since only one default export is allowed per module, this format is in fact semantically equivalent to the normalized format in Tab. 18 (b). Both formats demand the models to build a single module implementing all expectations, with a single default export. To help readers understand related concepts, we explain JavaScript export rules in Tab. 20.

Tab. 21 collects different ways models cope with this challenge. Tab. 21 (d) is the only right answer, but also the least straightforward, challenging the intuition trap that two exports from two separate

Table 20: Illustration of JavaScript default export in comparison to named imports

|  | **Named Exports** | **Default Export** |
|---|---|---|
| **Purpose** | Export multiple items from a module | Export a single item from a module |
| **Syntax** | `export const x = ...;`
`export function y() {...}` | `export default ...;` |
| **Import Syntax** | `import { x, y } from`
`'./module';` | `import anyName from`
`'./module';` |
| **Curly Braces** | Required during import | Not required during import |
| **Import Naming** | Must use the exact exported names (can use `as` to rename) | Can be imported with any name |
| **Multiplicity** | Multiple named exports per module | Only one default export per module |
| **Use Case** | Utility functions, constants, classes | Main functionality of a module |
| **Export Location** | Anywhere in the module | Bottom or after the main logic |

modules are needed. Both non-reasoning and reasoning models fall for the trap and attempt to split the implementation into two modules, (Tab. 21 (a), (b), (c)), resulting in very high failure rates.

Table 21: Patterns to address the duo-feature benchmark raw format (Tab. 18 (a))

```
function TaskA() {
  // Implementation of TaskA
}

function TaskB() {
  // Implementation of TaskB
}
export default TaskA;
export { TaskB };
```

(a) One default export and one named export

```
function TaskA() {
  // Implementation of TaskA
}

function TaskB() {
  // Implementation of TaskB
}

export { TaskA, TaskB };
```

(b) Two named exports

```
function TaskA_or_B() {
  // Implementation of TaskA or TaskB
}

export default TaskA_or_B;
```

(c) Only one task is implemented and exported

```
function TaskA_or_B() {
  // Implementation of both TaskA and TaskB
}

export default TaskA_or_B;
```

(d) Two tasks jointly implemented and exported

Next, we try to understand why non-reasoning models occasionally succeed by following the pattern of Tab. 21 (d), but non-reasoning models never do so. We suspect that the normalized format (Tab. 18 (b)) definitely dominates the pretraining/posttraining dataset, but does not exclude the raw format (Tab. 18 (a)), as well as the matching solutions. This makes the success possible.

On the other hand, from the first reasoning step which often plays the role of planning, reasoning models commit to the wrong judgment, and do not get a chance to correct the course in subsequent steps. Below is the detailed wording of the first reasoning step from a ChatGPT reeactment.

> To progress, the key task is creating components TaskA and TaskB in TaskA_B.js to ensure all tests are successfully passed.

Comparing to the mistakes made in Sec. E.1, the mistake in the above step covers a larger scope. It is reasonable to argue that mistakes made in large-scoped steps are more fatal and harder to correct.

## F  LINE-OF-CODE (LOC) ANALYSIS

Since top LLMs with SOTAs are proprietary, mechanistic studies are impossible. Therefore, we can only seek insights from model outputs. Thanks to the modularized design of the React framework, the solutions output by all models universally follow the template outlined in Tab. 3, with no need

for any explicit prompting. As such, we use LOC (line-of-code) as the proxy signal. Results in this appendix are from the single-feature benchmark.

## F.1 LOC DISTRIBUTION BY MODELS

Table 22: Models ranked by median LOC with pass@1

| Model | Median LOC | pass@1 |
|---|---|---|
| mixtral-8x7b-instruct | 35 | 0.1269 |
| llama-v3-8b-instruct | 39 | 0.0679 |
| llama-v3p1-405b-instruct | 40 | 0.3020 |
| gpt-4o-2024-08-06 | 40 | 0.8850 |
| deepseek-coder-v2-instruct | 40 | 0.7002 |
| gpt-4o-mini | 40 | 0.8271 |
| mistral-large-2 | 41 | 0.7804 |
| gemini-1.5-flash | 41 | 0.5700 |
| llama-v3p1-8b-instruct | 42 | 0.2512 |
| mixtral-8x22b-instruct | 43 | 0.3074 |
| claude-3.5-sonnet | 43 | 0.8808 |
| llama-v3-70b-instruct | 43 | 0.3323 |
| deepseek-coder-v2-lite-instruct | 43 | 0.4606 |
| gemini-1.5-pro | 45 | 0.6813 |
| llama-v3p1-70b-instruct | 46 | 0.1027 |

In Tab. 22, we rank models by their median LOC alongside their respective $pass@1$ scores. Picking one $pass@k$ is sufficient because all scores produced basically the same model rankings as shown in Tab. 6.

We observe that the median LOCs across all models stay close, ranging from 35 to 46. We believe this narrow range is largely enforced by the conciseness and expressiveness of the React framework itself. Also there is no strong correlation between the conciseness (median LOC) and correctness ($pass@1$). For example, mixtral-8x7b-instruct, which has the shortest median LOC, ranks quite low on $pass@1$ (0.1269). Conversely, stronger models like claude-3.5-sonnet and gpt-4o-2024-08-06, generate longer code. Other models, e.g. deepseek-coder-v2-instruct and gemini-1.5-pro, strike a balance between median.

Next, we use violin charts to visualize LOC distribution of each model. The distributions are either bimodal or unimodal, and they are collected in Fig. 5 and Fig. 6 respectively.

Notably, all high-performing models with high $pass@1$ scores are located in Fig. 5. These models, such as the gpt-4o variants and deepseek-coder series, demonstrate higher variability in their LOC distributions, i.e. bimodal. The two distinct peaks in these models' distributions suggests that they generate both shorter and longer code lengths, depending on the task. Importantly, the median LOC values for these bimodal models consistently fall between the two peaks, highlighting a balance in their code generation. Also the higher of the two peaks often corresponds to smaller LOC. This suggests that while these models can produce longer code when necessary, they tend to generate shorter, more optimized code in most cases.

In contrast, Fig. 6 contains smaller models. Some exhibit near-perfect normal distributions, e.g. mixtral-8x7b-instruct and llama-v3-8b-instruct. These models generate LOC distributions that are tightly centered around their medians, indicating more consistent and predictable behavior. The lack of bimodal characteristics in these distributions reflects a more stable output across tasks, but with lower complexity compared to the larger models in Fig. 5.

## F.2 IMPACT OF SUCCESS/FAILURE

To get more insights, we search for statistical distinction between successful model outputs and failed outputs. In Fig. 7 and 8, we visualize the LOC distribution separately for succssful outputs and failed ones, for each model. The graphs are ranked by $pass@1$, where higher $pass@1$ means

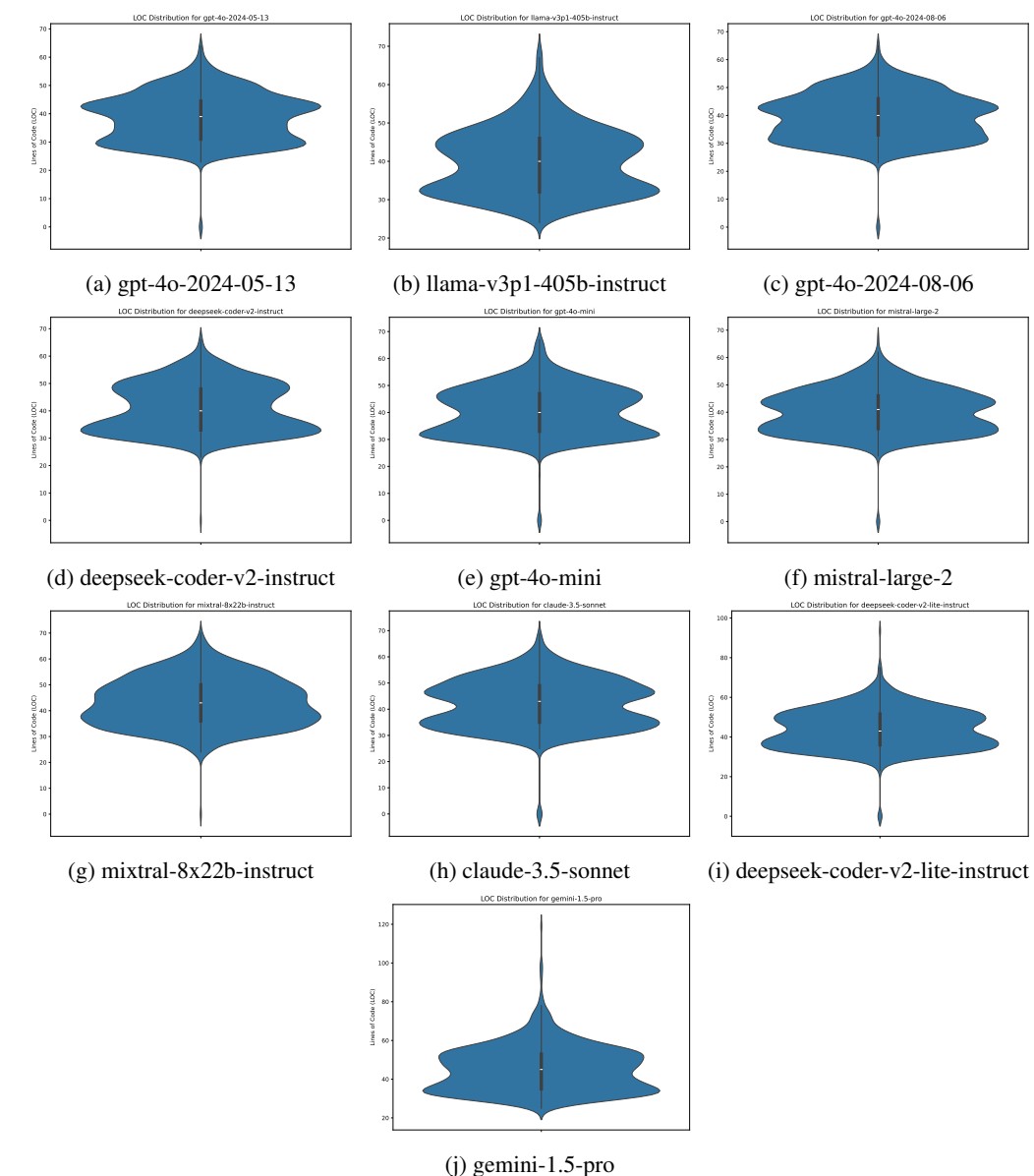

Figure 5: LOC distribution by model (bimodal)

bigger success sample set and smaller failure sample set. We normalize the width of each violin chart by its sample set size, hence resulting in the thinnest failure graph for the model with the highest $pass@1$. The graph gradually grows wider as the model performance degrades. The opposite pattern is observed for the success violin chart.

An important finding here is that the success distribution is always more complex than its failure counterpart, with more peaks and deviations. Fig. 8 groups lower performing models whose failure sample set dominates the success sample set. The failure LOC distributions are unimodal, in contrast with the multimodal distributions of top models in Fig. 7. This implies the inherent complexity involved in writing correct code even when the mean LOC is less than 50.

The success/fail LOC distribution of remaining 8 models are shown in Fig. 9.

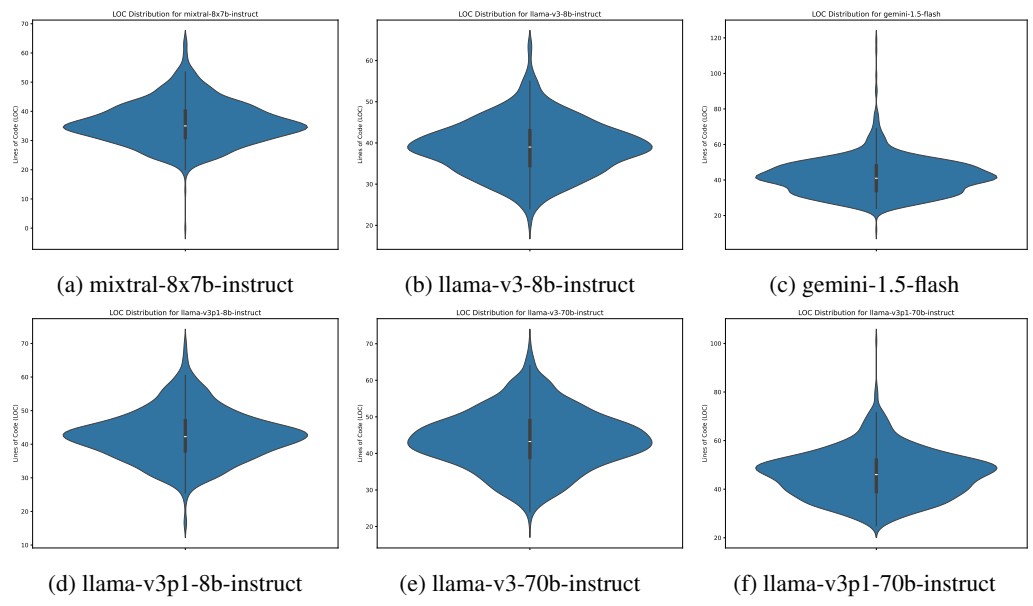

(a) mixtral-8x7b-instruct     (b) llama-v3-8b-instruct     (c) gemini-1.5-flash

(d) llama-v3p1-8b-instruct     (e) llama-v3-70b-instruct     (f) llama-v3p1-70b-instruct

Figure 6: LOC distribution by model (unimodal)

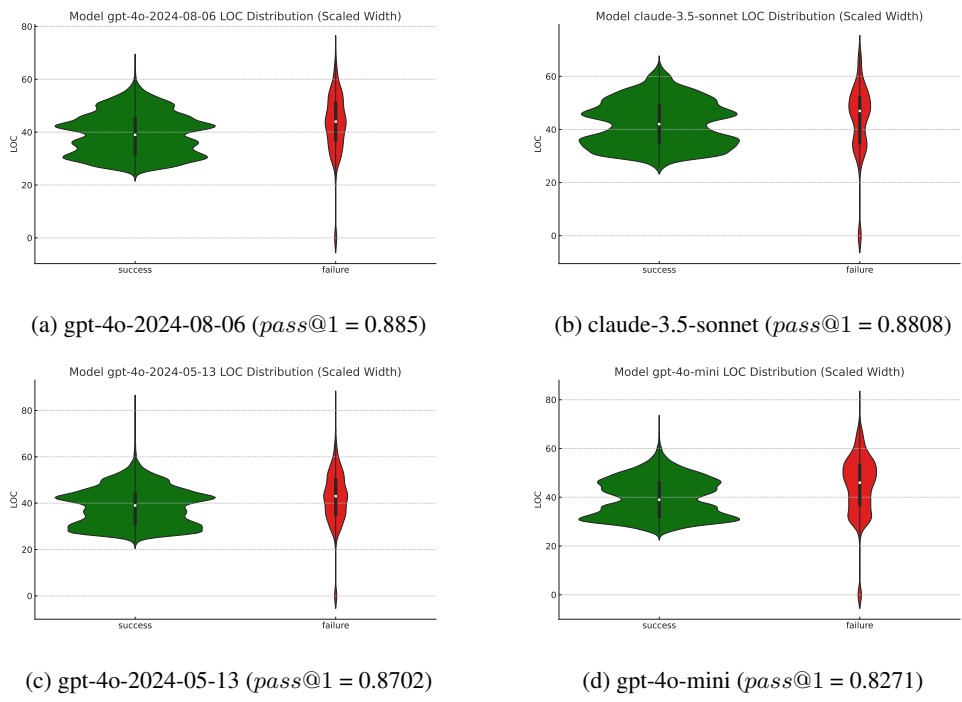

(a) gpt-4o-2024-08-06 ($pass@1 = 0.885$)     (b) claude-3.5-sonnet ($pass@1 = 0.8808$)

(c) gpt-4o-2024-05-13 ($pass@1 = 0.8702$)     (d) gpt-4o-mini ($pass@1 = 0.8271$)

Figure 7: LOC distribution by model of high pass@1: success vs failure

## F.3 LOC DISTRIBUTION BY APPLICATIONS

In Tab. 23, we rank median LOC for each application. Consistent with the case for model ranking (Tab. 22), the median values stay within a narrow range (37 to 46). This suggests that all models consistently produce solutions of similar length, irrespective of the task complexity or domain.

Fig. 10 collects violin charts of 14 applications following unimodal distribution, where the model outputs are centered around a common length, with less variation between extremes. The remaining

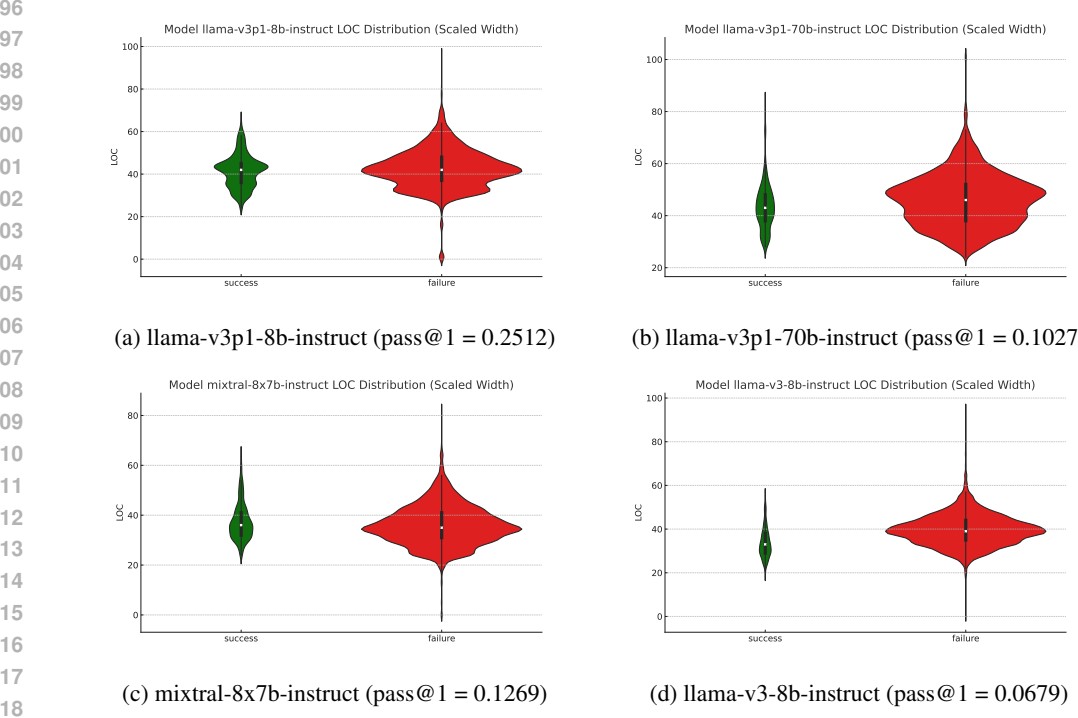

(a) llama-v3p1-8b-instruct (pass@1 = 0.2512)  (b) llama-v3p1-70b-instruct (pass@1 = 0.1027)

(c) mixtral-8x7b-instruct (pass@1 = 0.1269)  (d) llama-v3-8b-instruct (pass@1 = 0.0679)

Figure 8: LOC distribution by model of low pass@1: success vs failure

Table 23: Applications ranked by mean LOC

| Application | Mean LOC |
|---|---|
| News Aggregator | 37 |
| Music Streaming | 37 |
| Online Marketplace | 37 |
| E-commerce | 37 |
| Recipe Sharing | 38 |
| Fitness Tracking | 38 |
| Online Learning | 38 |
| Blogging | 39 |
| Weather | 40 |
| Real Estate | 42 |
| Social Media | 42 |
| Job Board | 42 |
| Inventory Management | 42 |
| Pet Care | 42 |
| Travel Planning | 42 |
| Personal Finance | 43 |
| Customer Support | 44 |
| Photo Gallery | 44 |
| Event Management | 45 |
| Task Management | 46 |

6 applications are in Fig. 11, following multimodal distribution. In both cases, the median LOC is always positioned centrally in each distribution, which suggests that the code generation is stable across applications. Applications in Fig. 11 exhibit more complex patterns, but the distributions remain balanced with the median value positioned at the center of the distribution.

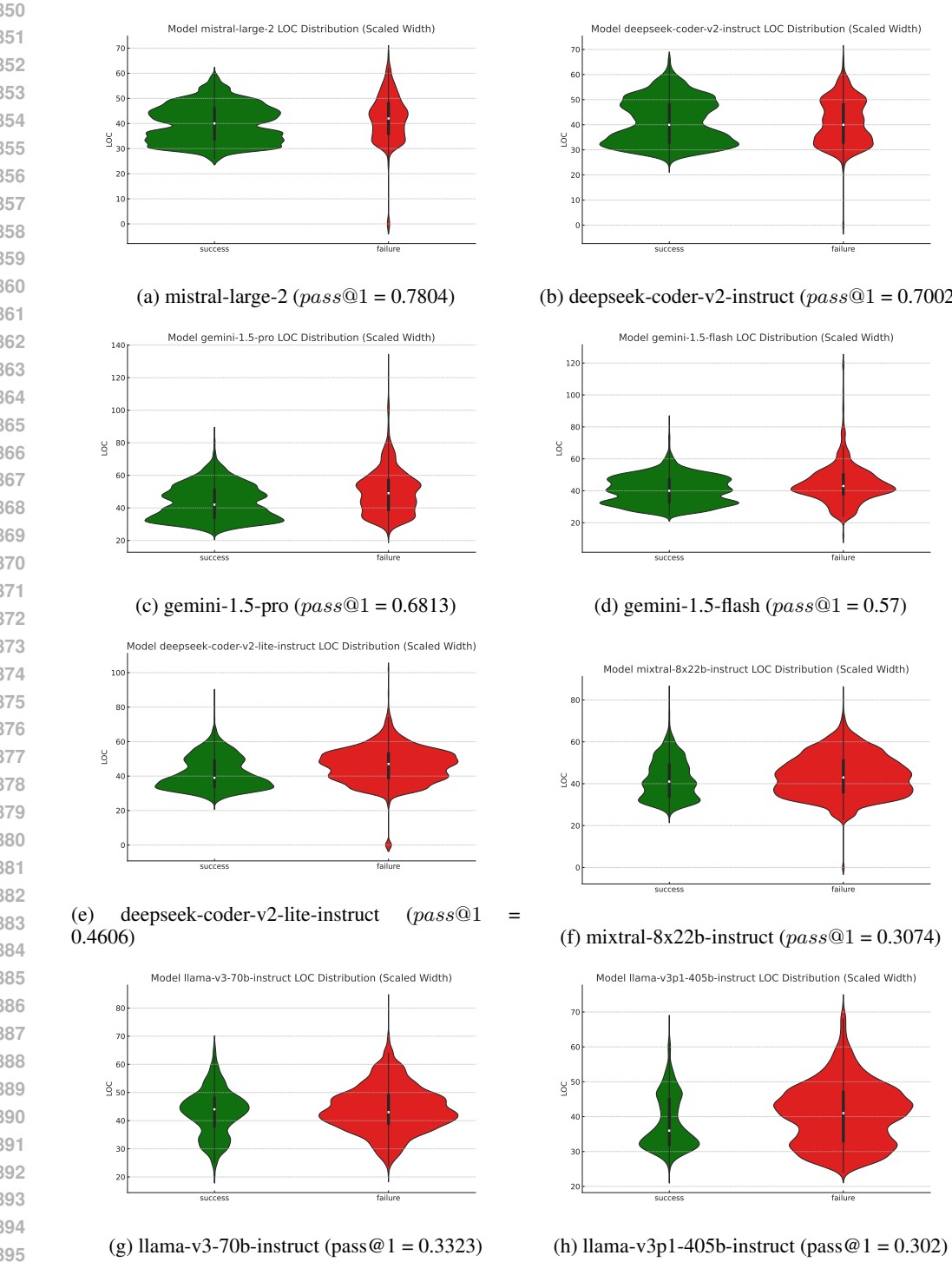

(a) mistral-large-2 ($pass@1 = 0.7804$)

(b) deepseek-coder-v2-instruct ($pass@1 = 0.7002$)

(c) gemini-1.5-pro ($pass@1 = 0.6813$)

(d) gemini-1.5-flash ($pass@1 = 0.57$)

(e) deepseek-coder-v2-lite-instruct ($pass@1 = 0.4606$)

(f) mixtral-8x22b-instruct ($pass@1 = 0.3074$)

(g) llama-v3-70b-instruct (pass@1 = 0.3323)

(h) llama-v3p1-405b-instruct (pass@1 = 0.302)

Figure 9: LOC distribution by model: success and failure

## F.4 LOC DISTRIBUTION BY APPLICATIONS: SUCCESS VS FAILURE

We conduct the same study described in Sec. F.2, except we shard the LOC distribution across applications instead of models. The results are collected in Fig. 12.

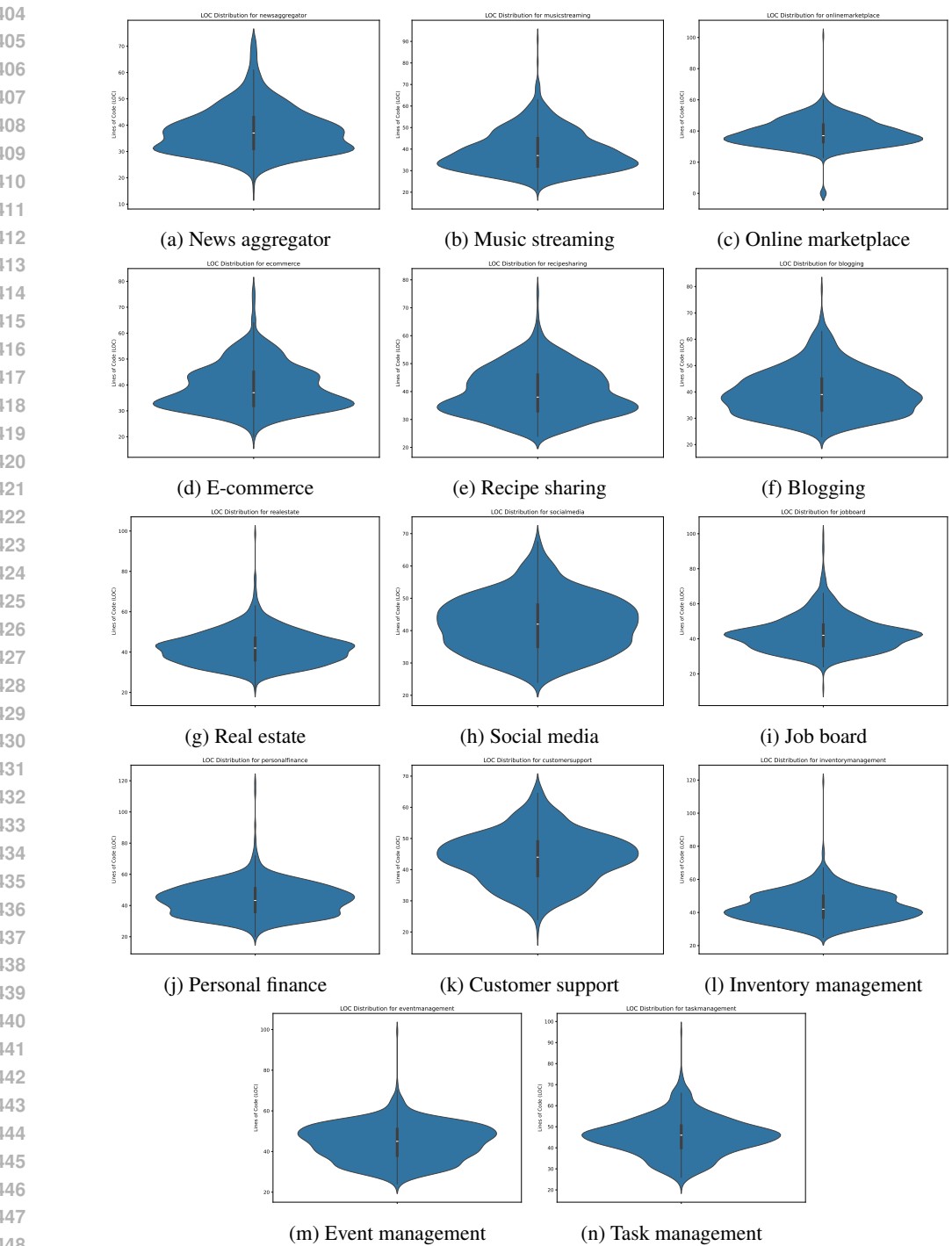

Figure 10: LOC distribution by applications: unimodal

Since each application assembles outputs from all models with full spectrum of performances, the success and failure data set are about the equal size. Similar to what we have observed in model-based sharding (Sec. F.2), the distribution pattern for success is equally or more complex than that for failure, summarized in Tab. 24.

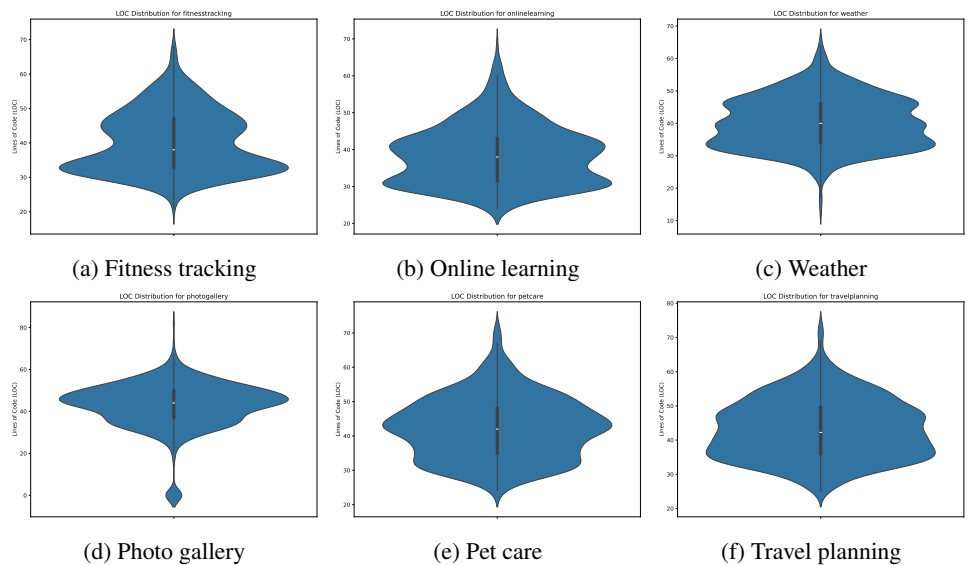

Figure 11: LOC distribution by applications: multimodal

Table 24: Summary of Fig. 12: unimodal vs multimodal

|  | **UniModal Success** | **MultiModal Success** |
|---|---|---|
| **UniModal Failure** | (b) (q) (t) | (c) (d) (f) (g) (h) (j) (k) (l) (m) (n) (o) (p) |
| **MultiModal Failure** |  | (a) (e) (i) (r) (s) |

## G   PER-APPLICATION ERROR ANALYSIS

Fig. 13 shows the failure pattern broken down by applications.

1. *Consistency Across Applications*: All applications exhibit the same general shape—a large concentration of easier problems on the left side and a few harder problems on the right side. This consistency suggests that across different domains, there are always a few particularly challenging problems that models struggle with.

2. *Variations in Skewness*: Some applications, such as Fitness Tracking and Music Streaming, show a more pronounced skew with a sharp rise in failure rates for a few problems, indicating a steeper difficulty curve. Others have a more gradual increase, indicating a more even distribution of problem difficulty.

3. *Extreme Difficulty in Certain Applications*: Applications like Customer Support and Pet Care have a sharper increase towards the right, implying that these domains have a subset of problems that are especially challenging.

4. *Easier Applications*: In applications like Weather and Photo Gallery, the overall number of failures seems lower compared to other applications, suggesting that the problems in these areas were generally easier.

Fig. 14 shows error distribution by applications. Since each application assembles outputs from all models, the raw error counts are at the same scale for all applications. We do not find any distinctive patterns. There is neither special error nor special application.

## H   BIAS ANALYSIS

We conducted a preliminary investigation into potential biases within our benchmark, focusing on language bias, cultural inclusivity, and implicit assumptions. To this end, we searched the codebase for gendered terms, stereotypical language, and regional references using an automated analysis

script. Additionally, we examined API endpoints and user-facing messages for exclusionary patterns or implicit biases. Our investigation did not identify any instances of such biases in the current version of the benchmark.

While these findings are encouraging, we recognize the limitations of automated analysis and the potential for more nuanced biases that may require further investigation. We welcome additional guidance or suggestions for extending this analysis to ensure a comprehensive evaluation of fairness within our benchmark.

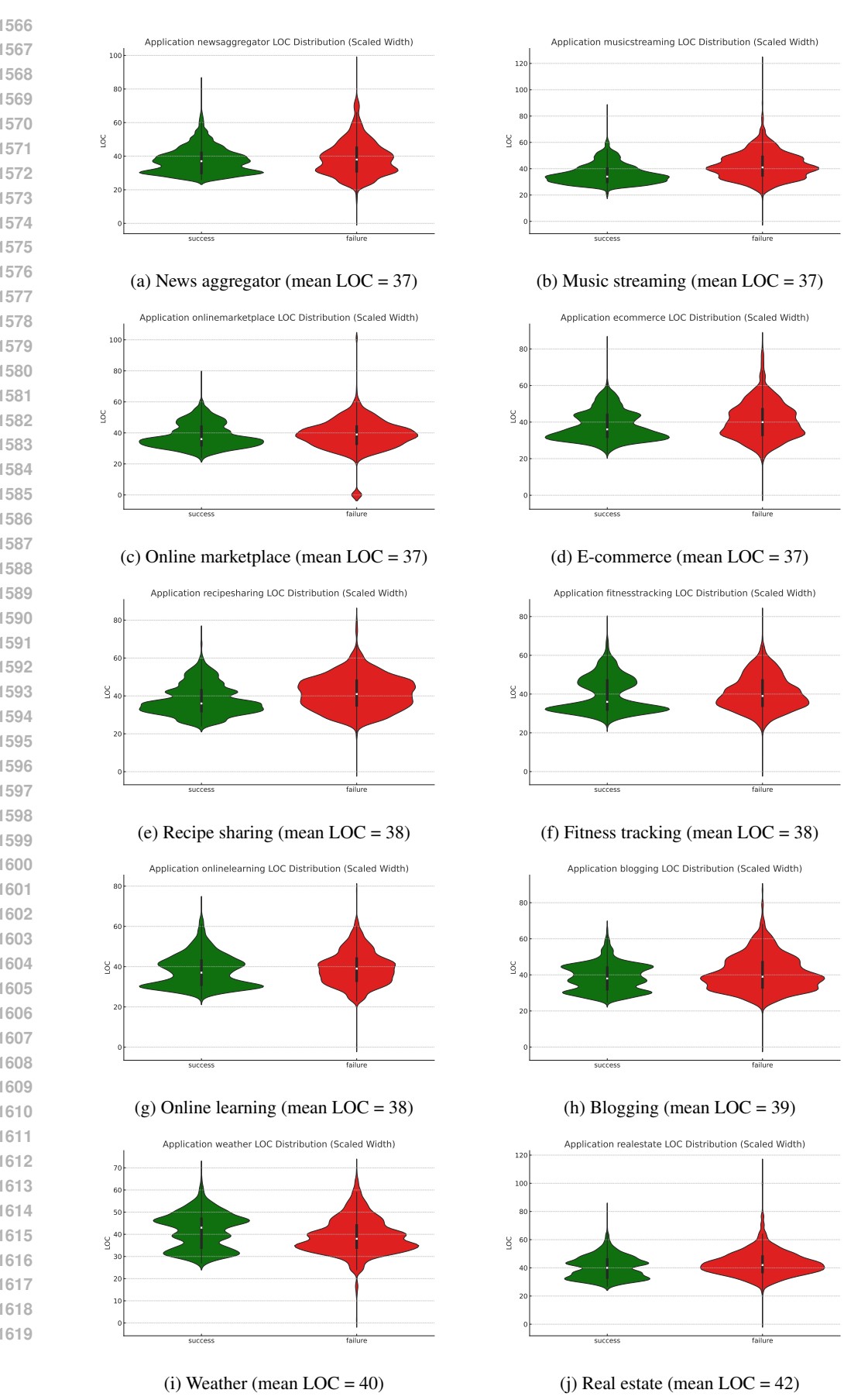

(a) News aggregator (mean LOC = 37)

(b) Music streaming (mean LOC = 37)

(c) Online marketplace (mean LOC = 37)

(d) E-commerce (mean LOC = 37)

(e) Recipe sharing (mean LOC = 38)

(f) Fitness tracking (mean LOC = 38)

(g) Online learning (mean LOC = 38)

(h) Blogging (mean LOC = 39)

(i) Weather (mean LOC = 40)

(j) Real estate (mean LOC = 42)

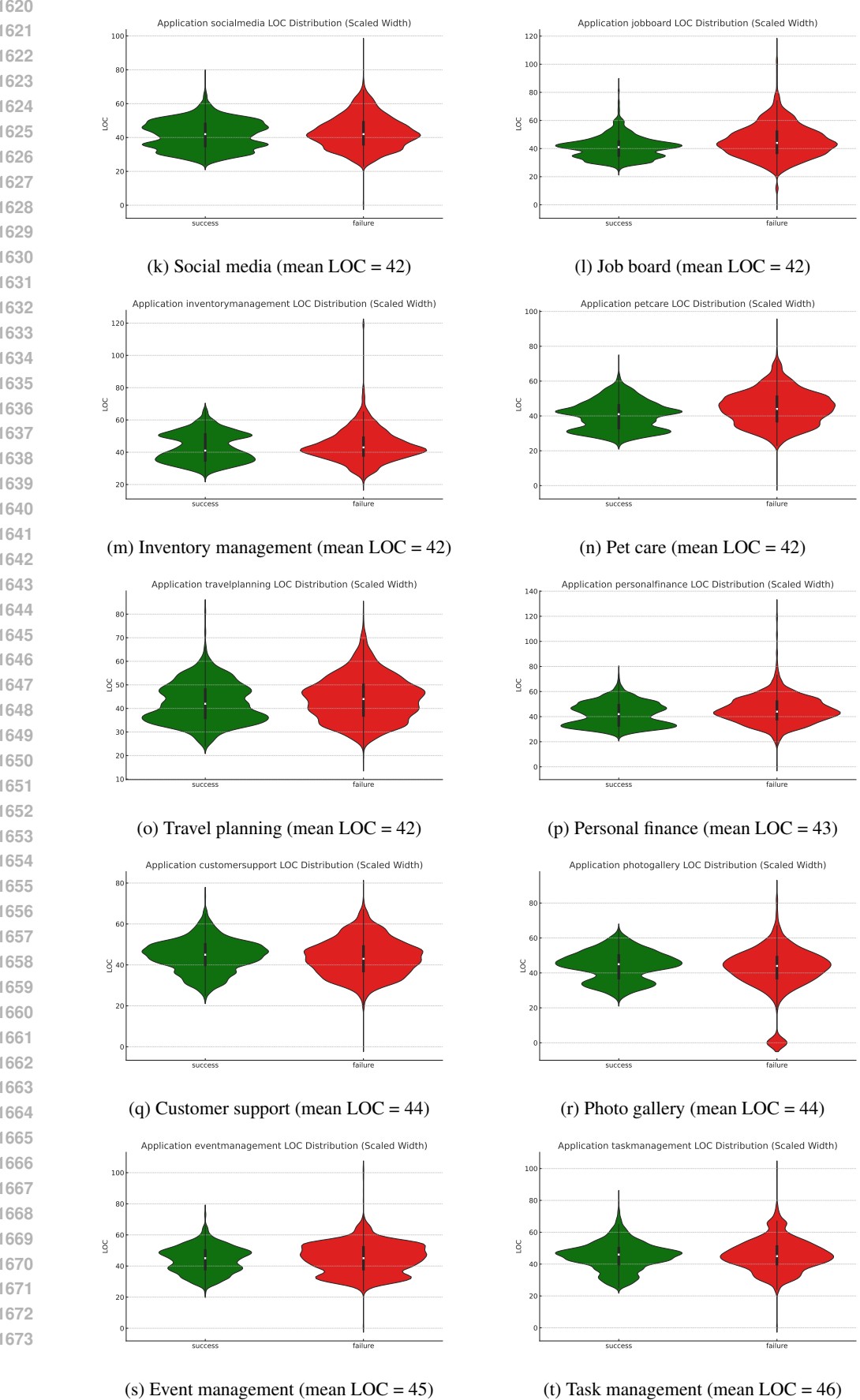

(k) Social media (mean LOC = 42)

(l) Job board (mean LOC = 42)

(m) Inventory management (mean LOC = 42)

(n) Pet care (mean LOC = 42)

(o) Travel planning (mean LOC = 42)

(p) Personal finance (mean LOC = 43)

(q) Customer support (mean LOC = 44)

(r) Photo gallery (mean LOC = 44)

(s) Event management (mean LOC = 45)

(t) Task management (mean LOC = 46)

Figure 12: LOC Distribution by Application: Success vs Failure

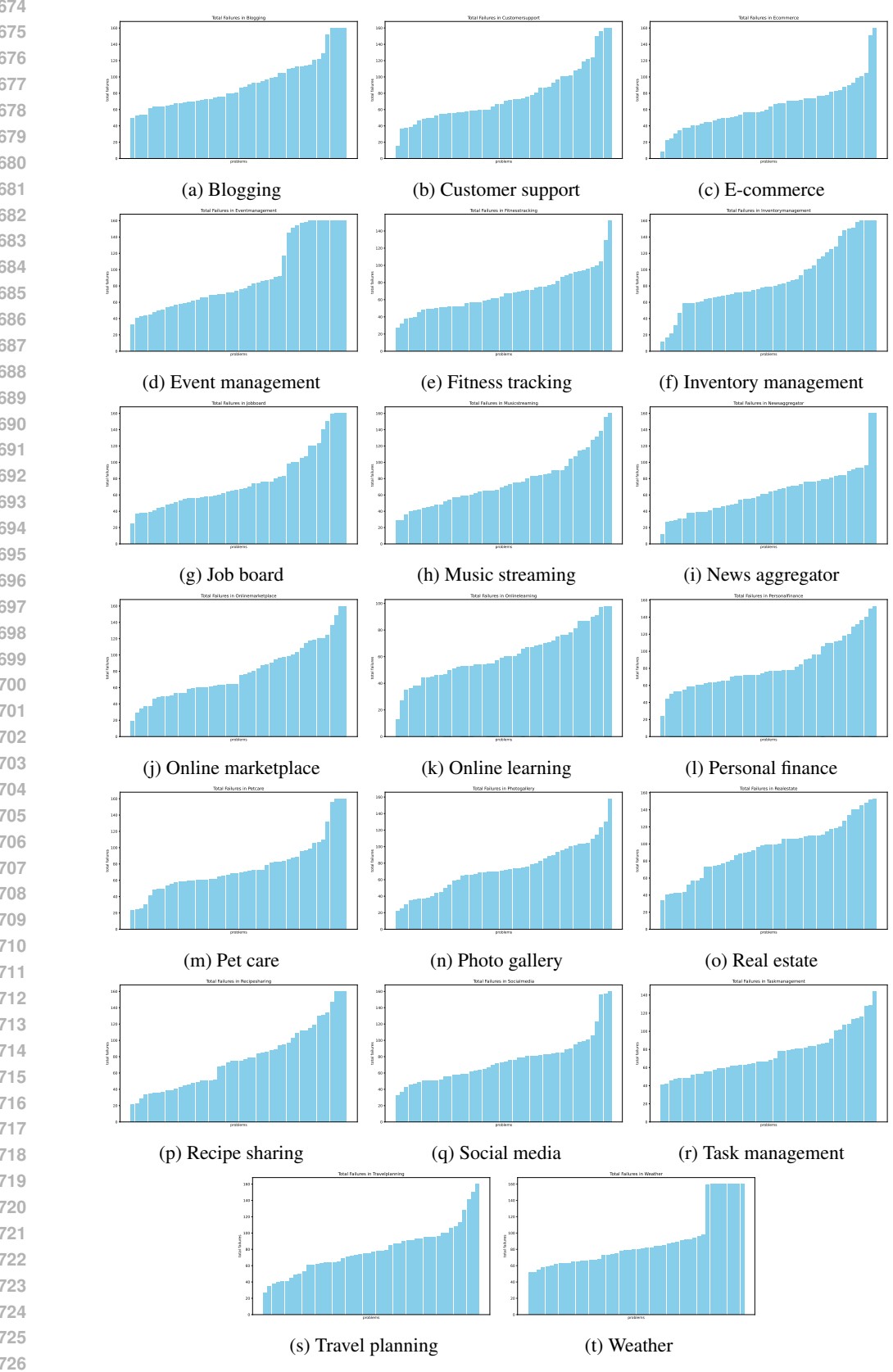

Figure 13: Failures per problem by application

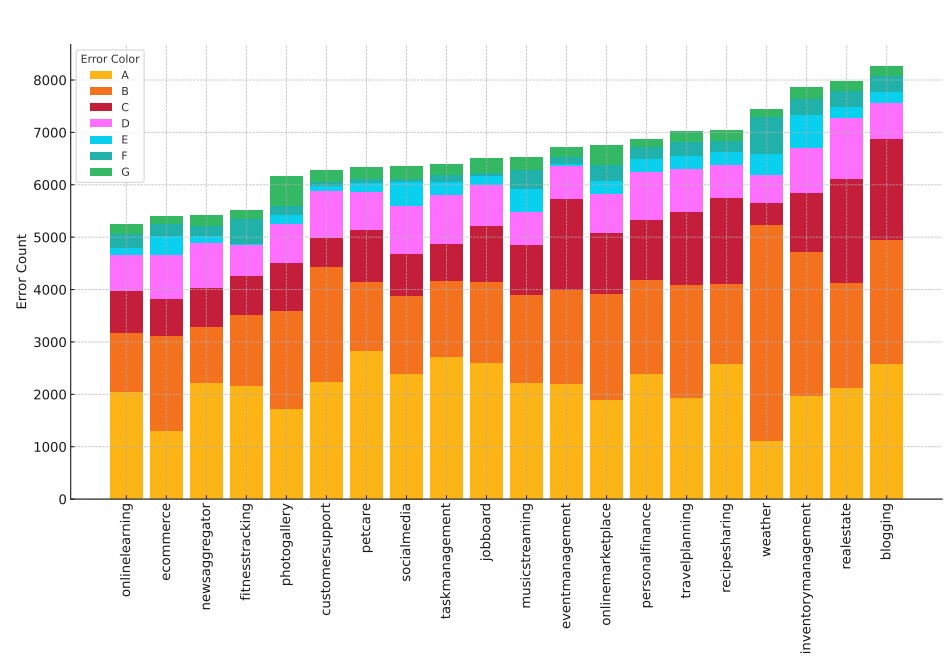

Figure 14: Errors by applications

