# OpenReview forum: "Tests as Instructions: A Test-Driven-Development Benchmark for LLM Code Generation"
_ICLR.cc/2025/Conference — Submitted to ICLR 2025_

### Official Review · Reviewer_7wcY · 2024-11-02

**Soundness:** 2
**Presentation:** 1
**Contribution:** 2
**Rating:** 3
**Confidence:** 4

**Summary:**

This paper explores test-driven development (TDD) tasks, utilizing test cases as both guidance and verification mechanisms for LLM-based code generation. The authors introduce a TDD benchmark specifically designed to evaluate the performance of 18 leading-edge LLM models, with OpenAI’s reasoning models achieving state-of-the-art results. The paper highlights two main contributions: the development of a specialized TDD benchmark for assessing LLMs and the insights derived from this evaluation, notably that instruction following and in-context learning are critical for improving LLM performance on complex TDD tasks.
The review requires several clarifications from the authors. Firstly, the paper indicates the presence of a benchmark comprising 1000 tasks (line 106), yet it lacks a comprehensive statistical analysis and comparison concerning its data sources, including the balance of data distribution and the types of data used. Given the emphasis on this benchmark as a significant contribution, a detailed description of the data statistics, collection processes, and methodologies is essential to assess any potential biases.

The introduction could be clearer, particularly the sections detailing the motivation behind the study. Including a concrete example of the TDD task and its motivations would enhance understanding. It would also be beneficial to concisely summarize the paper's contributions in the introductory section.

The presentation of the tables and figures needs improvement for better clarity. For instance, employing different color fonts or using symbols like boxes to highlight critical code segments in Tables 1, 2, and 3 would aid comprehension, especially for readers unfamiliar with specific coding frameworks such as React. Additionally, it is unclear what the horizontal axis represents in Figure 1.

In the section on evaluation results, the paper should address several aspects to provide a more comprehensive presentation of the findings. These include the rationale behind choosing pass@k as an evaluation metric, whether this metric alone suffices to demonstrate the capabilities of LLMs, the setup for the evaluations, and how these evaluations compare with existing benchmarks.

The term 'incomplete manual examination' mentioned in line 225 is vague. Clarification is needed on how many solutions were manually inspected and whether this sampling is sufficient to support the conclusions drawn.

Lastly, the evaluation results section appears overly verbose. Streamlining this section would make the paper more concise and better aligned with the expectations of a research publication, rather than resembling a lab report.

**Strengths:**

new benchmark development;
Detailed Experiments;
Relevance to Current Challenges

**Weaknesses:**

Lack of Comprehensive Data Analysis;
Clarity in Presentation

**Questions:**

1. Can you provide more comprehensive details about data statistics, collection processes, and methods to evaluate potential biases in the benchmark?

2. Could you include a concrete example to better explain the TDD task and its motivations? Also, could you summarize the contributions of the paper in the introduction for clearer presentation?

3.  Why was pass@k chosen as the evaluation metric, and do you believe one metric is sufficient to demonstrate the capabilities of LLMs? Could you also detail the evaluation setup and compare it with existing benchmarks?

4. What does "incomplete manual examination" mean?

---

> ### Author Response · Authors · 2024-11-22
>
> Thank you for the review and comments. We upload a revision with rewritten introduction, and multiple appendices documenting details of our experiments.
>
> Q1: Data Analysis and collection processes
>
> We added several appendices. Specifically, construction of the benchmark (Appendix A), Test-last Development (TLD) experiment (Appendix C), line-of-code (LOC) analysis (Appendix F), per-application error analysis (Appendix G).
>
> Q2: Explanation of TDD and contribution clarifications
>
> We rewrote the introduction to include a TDD example referencing a classical TDD book, and clarified contributions.
>
> Q3: Bias
>
> We searched biased terms (social status, gender, regional, stereotypical) in our benchmark and model outputs and found none. More advice is appreciated. We also attach the benchmark source code as supplementary materials.
>
> Q4: pass@k as the sole evaluation metric
>
> pass@k is the de facto leaderboard metric widely accepted by the community. So we believe it will stay dominant in the near future. Also worth noting is the inclusiveness of pass@k. If a certain criterion is crucial and objective, then we can codify it into an automated test, then add to the benchmark (or fork a new benchmark). For example, a bias-aware benchmark can include a bias filtering test, hence still use pass@k.
>
> A main drawback of pass@k is that each instance only produces a binary value, not numeric value for usefulness. Our TLD study shows that a model can output code that is functional but not up to the testing spec. These are important supplementary signals to support pass@k, along with analytical signals such as LOC and error types.
>
> Q5: Comparison with existing benchmarks
>
> We added a comparison table in Introduction to compare TDD benchmarks with algorithmic and problem-solving benchmarks such as SWE-bench. Also experimental setup is described in Appendix B.
>
> We try to make our benchmark as lightweight as algorithmic benchmarks, and hope other benchmark builders will agree with us. Our desired goal is to equip model trainers with more benchmarks for interim evaluation when their models are still in the making. If the benchmarks can help point hill-climbing directions, then it’d be even better.
>
> Q6: incomplete manual examination
>
> What we did is manually examine model outputs to assess the code usefulness, i.e. is the code still functional even if they break tests. Given the large volume of model outputs, we can only sample a very small subset. We automated this process via the TLD experiment (Appendix C).

---

> > ### Comment · Reviewer_7wcY · 2024-11-27
> >
> > Thank you to the author for their reply. While I partially agree with the importance of the topic addressed in the paper,  the writing of this paper still requires significant improvement. Additionally, the benchmarks presented so far lack sufficient public cases and details. Therefore, I will keep my score unchanged.

---

### Official Review · Reviewer_6weu · 2024-11-03

**Soundness:** 1
**Presentation:** 1
**Contribution:** 1
**Rating:** 1
**Confidence:** 4

**Summary:**

This paper proposes a test-driven development (TDD) benchmark, where the model needs to generate code to pass given test cases. The benchmark consists of 1000 Javascript-based web app development tasks. This paper evaluates modern LLMs' performance on the proposed benchmark (e.g., o-1, claude, deepseek, etc). Error analyses show that most LLMs can generate syntax-error-free code, and often pass at least one test case. The paper also proposes a more challenging benchmark by merging two original TDD tasks into a duo-feature task.

**Strengths:**

1. TDD is an important evaluation task that meets real-world coding scenarios.

**Weaknesses:**

1. Limited test cases are shown to be hacked by LMs, i.e., LMs often generate incorrect code solutions that can still pass these test cases.
2. Lack of benchmark details. In line 106, the paper claims the benchmark consists of 1000 examples, while never clarifying how these examples are collected.
3. Poor paper writing. All references and tables are in incorrect format. There are a lot of redundant sentences (e.g., in line 161, since $k$ most be smaller or equal to $n$). Duplicated contents (e.g., both Section 2.2 and Section 4.1 are named Task Formulation). Weird paper structure (e.g., Section 4 is merely a case study of o1.

**Questions:**

see weakness above

---

> ### Author Response · Authors · 2024-11-22
>
> Thank you for the review and comments. We upload a revision with rewritten introduction, and multiple appendices documenting details of our experiments.
>
> Q1: LM hacking
>
> We fully acknowledge the risk of false positives, i.e. incorrect solutions passing the tests. Our choice of React mitigates the risks at a certain level. The code is highly templated because low-level details are abstracted away to the framework. We manually compare model outputs and find them to be highly similar. Also in our future work, we plan to add non-unit tests, e.g. integration and accessibility tests to further mitigate the risk.
>
> Q2: Lack of benchmark details.
>
> We added benchmark construction and experiment details to Appendices A and B.
>
> Q3: Poor paper writing.
>
> We rewrote the Introduction section, moved o1 discussions to the appendix.We also reformatted table and figure. Our paper used the iclr2025 template, and the references used \citep. More advice is appreciated.

---

### Official Review · Reviewer_2tsP · 2024-11-04

**Soundness:** 3
**Presentation:** 3
**Contribution:** 3
**Rating:** 6
**Confidence:** 4

**Summary:**

This paper introduces a new test-driven-development benchmark for code LLMs, wherein the instructions given to the language model is the code of the test directly, without going through a natural language description of the task. A benchmark of 1000 React tasks is constituted.

18 models, proprietary and open-source, of various sizes, are tested on the benchmark. The authors analyze the failure types and propose explanations for the various types of failures. They find that the latest o1 models from OpenAI, trained to reason in natural language before committing to an output, achieve impressive scores on the benchmark but forget part of the instruction, leading to lower scores than Sonnet on the most complex tasks.

**Strengths:**

## Originality

The paper presents the first fully-blown benchmark with test-as-instructions ([Programming Puzzles](https://arxiv.org/abs/2106.05784) also used code as instructions but was focused on algorithmic or mathematical puzzles). As argued in the paper, this prevents a gap between specification and test cases.

## Quality

The analyses in the paper are of good quality and many models are tested. The failure cases are analyzed by the authors, and concrete examples are given, which illustrates the problem and skills required to solve it.

## Clarity

The paper was straightforwardly written, and easy to follow along.

## Significance

I could see this benchmark being used to assess programming abilities of LLMs or LLM-based systems. As outlined by the authors themselves, test-driven development looks more similar to the way software is developed in teams, and could lead to studies for products that could help developers' productivity.

**Weaknesses:**

* I am concerned that the benchmark is too easily saturated by models already. A fully blown, public release of this benchmark should use the multi-feature mode that makes tasks more difficult; and probably use some form of validation (having software engineers write code) to make sure that the tests for the multiple features are indeed satisfiable;
* Another limitation of the paper is that it does not discuss the differences between it and SWE-bench-verified, which has become a standard for coding LLMs. More discussion should be included (for instance following the discussion in 6.4)
* A final concern is the data collection, which is left in the dark in this paper. Where do the tests come from? Are they scraped from Github, from learning websites? How close to the pretraining data are these tasks? Could contamination explain the really high scores?

As a final note I would like to add that I like this paper and I would be willing to raise my score if my concerns (especially on construction of the dataset) are answered.

**Questions:**

* Have you tried running the benchmark with various forms of agents, and execution feedback? This should easily correct some of the error categories you have discovered (such as version mismatch or uninstalled module);
* Can you provide additional visualizations, explanations and statistics on the contents of the dataset? Perhaps a few additional examples in the appendix as well would be useful.

---

> ### Author Response · Authors · 2024-11-22
>
> Thank you for the review and comments. We upload a revision with rewritten introduction, and multiple appendices documenting details of our experiments.
>
> Q1: Benchmark saturation
>
> We agree benchmark saturation is a very important issue. Our original design objective to identify performance bottlenecks when required skills and knowledge are already in the pretrain data. We are confident that React is sufficiently represented in pretrain data of all models. So the saturation of top models is somewhat expected. But the first unanswered question was why weaker models perform so much worse than top models. After seeing performance drop of all models on duo-features, we turn to non-coding abilities, e.g. instruction following. We also ran a TLD (test-last development) experiment to show that weaker models actually write functional code, only not up to the test spec (Appendix C). We now suspect the root cause to be attention decay, i.e. the models need to attend to multiple needles in the haystack (the more tests, the more needles). But mechanistic studies are needed to confirm our suspect.
>
> We agree multi-feature is mandatory for the benchmark release. Each application in the benchmark consists of 50 features, which is the theoretical limit. As of now two features already lower SOTA significantly.
>
> Q2: Additional tests
>
> We are now evaluating tests to include in addition to unit tests, such as integration tests, accessibility tests, rendering tests, etc. But we do not have any results yet. A nice thing about TDD is its inclusiveness: you can add these tests to the prompt and reuse the benchmark.
>
> Q3: Human inputs
>
> Due to budget constraints, we haven’t hired human testers to write code. We plan to launch an arena (similar to LMSys) to invite users to rate our model outputs, and also share their own inputs.
>
> Q4: Difference to SWE-bench
>
> We added a comparison table in Introduction to compare our benchmark against SWE-bench. Basically our benchmark targets the head-down coding phase of the project, and SWE-bench is targeted for the post-launch phase. Also different to SWE-bench, we try to make our benchmark easy to light to run, such that model trainers can use it for interim eval before releasing their models.
>
> Given the proliferation of coding models and the declining training cost, we expect more companies to build their own specialized models (maybe with their own proprietary codebase), as well as their own benchmarks following our methodology.
>
> Q5: Data collection
>
> The benchmark is synthetic using seeded human inputs (following the self-instruct method). We reference React learning sites and books to decide which applications to benchmark. The test cases are written by GPT-4o. Since React is a highly prominent framework, the knowledge is contaminated in all pretrain corpora. As analyzed in the first question, we believe it’s non-coding abilities which differentiate model performances.
>
> Q6: Agents
>
> We did not build an agent, but tried several prompting techniques including retries (Appendix D). We also tried an open source CoT approach mimicking reasoning models. They didn’t help. As rationalized in the first question, we suspect the problem is rooted inherently in the model itself.
>
> Q7: More statistics and visualizations
>
> We added several appendices to explain the benchmark and experimental setup. We also conducted an extensive line-of-code analysis from model outputs. Our research focus is now shifted to open models because we must examine internal states to get deeper insights.

---

### Official Review · Reviewer_NQZp · 2024-11-04

**Soundness:** 3
**Presentation:** 3
**Contribution:** 2
**Rating:** 6
**Confidence:** 5

**Summary:**

This paper introduces a benchmark for evaluating LLM on test-driven development. The prompt simply contains a list of test-cases, and instructions to pass them, while returning only code. There are no hidden text-cases. To pass, LLM needs to generate code which passes all the tests in the prompt.
The benchmark contains 1000 React problems (this is a popular java script library) and solutions of 16 LLM to these problems, where each LLM generates response 10 times to estimate test failure rate. The benchmark contains a good distribution of task difficulties, from trivial to unsolvable. The authors categorize errors that cause test-cases to fail, into categories such as missing library import, deprecated frameworks are used, etc. The find that not surprisingly adding more test-cases to the prompt is more likely to cause failure (2 and 4 test-case prompts are evaluated), and that o1 models show superior performance.

I am not familiar with React, and not willing to learn it for the purpose of writing this review, so I just skipped most of the exampled.

**Strengths:**

I was looking forward to reading this paper, as the issue is certainly topical. Many of these benchmarks is being created, and this one is novel.

Benchmark generation is very costly, so the current project is useful.

**Weaknesses:**

Being focused on React is the main weakness. The authors claim that this is done because React is popular, and the code it produces is brief, but I am not convinced. Surely many python problems can be solved with few lines of code, and Python is a more popular language. This needs to be motivated.

Despite initial excitement, I feel like I learned very little from this paper. It seems obvious that most LLM coding mistakes come from missing libraries, ignoring instructions and training on legacy code. It would help focus presentation to clarify the conclusions.

There is no discussion of how adding more tests breaks LLM. For example, does the same happen to human coders? Adding TWO issues to implement in a prompt is not the same as making the prompt longer with 4 test-cases, please clarify the framing around this.

**Questions:**

What was the real reason for using React?

Performance seems to be very stable, such that LLM do not improve very much between pass@1 and pass@10. This seems interesting, but why is that?

---

> ### Author Response · Authors · 2024-11-22
>
> Thank you for the review and comments. We upload a revision with rewritten introduction, and multiple appendices documenting details of our experiments.
>
> Q1: Choice of React
>
> Popularity and conciseness of React help us evaluate LLMs as they are, i.e. limited context windows and  pretrained knowledge. Another practical consideration is the SPA (single-page application) property of React, i.e. you can concentrate all implementation of an app in one file. Controlling LLMs to generate multiple files is still a pain point, such that SWE-bench branched a SWE-bench-lite variant to focus on issues requiring one-file commit only.
>
> Having said this, Python Django is on our roadmap, but the code is spread to three files at least (model, view, controller). So extra steps are needed to force the model to generate one file only while anchoring the other two files.
>
> Finally, we advocate for fine-granularity benchmarks. The training cost of coding models drops so quickly, that we see a future where many companies ship their own open/proprietary models, along with their own benchmarks. So one can envision a new Python benchmark to evaluate Python-specialized coding models.
>
> Q2: Clarifying conclusions of the paper
>
> We rephrased our contributions in the introduction. To accentuate the challenge of TDD, we also designed a TLD (test-last development) experiment (Appendix C), i.e. modifying tests to accommodate model-generated code. Without a priori restrictions of tests, all models can generate functioning code with 60%+ pass@1. Software engineers can build demos using TLD, but have to follow TDD for table stake projects, where model performances fall short shown by our study.
>
> Q3: Impacts of more tests
>
> We suspect attention decay to be the root cause of weaker performance when more tests are added. This problem is akin to parameter parsing in function call and needle in haystack, in which there are multiple needles. The longer the tests, the more needles you need to find and keep. Through the TLD experiment, we show that all models have strong coding abilities, but we suspect insufficient attention is why they fail.
>
> This is also the reason we plan to shift research focus to open source models, not only to avoid high cost of proprietary models, but also to monitor internal states like activations and attention scores.
>
> Q4: Close performances between pass@1 and pass@10
>
> To ensure fairness, we grid searched unified sampling parameters for all models to follow (Appendix B). The chosen parameters (e.g. temperature 0.2) are biased towards more deterministic outputs. Also if attention decay is confirmed to be the performance bottleneck, then sampling (more evals or more diversified outputs) is probably too late to help.

---

> > ### Comment · Reviewer_NQZp · 2024-12-01
> > **Thanks for your reply.**
> >
> > I have read the comments and thank the authors for their detailed response.

---

### Meta-Review · Area_Chair_dqtQ · 2024-12-21

**Metareview:**

This paper introduces a benchmark for evaluating LLM on test-driven development. Code generation is definitely an important question for LLM. The concerns are three-folded. First, is it already over-saturated for the current advancement of LLM. Second, what is its unique add-on compared to other code generation benchmarks (e.g., SWE-bench). Third, the experiment results are only run with ReAct, which leaves the results questionable for its general applicability. Furthermore, due to its unclarity of writing, it is difficult to objectively evaluate its true merit.

**Additional Comments On Reviewer Discussion:**

The main sharing concern regarding the clarity and writing of this paper still remains after the rebuttal.

---

### Decision · Program_Chairs · 2025-01-22

Reject